

**Role of thermodynamic and turbulence processes on the fog life cycle during**
**SOFOF3D experiment**
Cheikh DIONE[1*], Martial HAEFFELIN[1], Frederic BURNET[2], Christine LAC[2], Guylaine CANUT[2], Julien
DELANOË[3], Jean-Charles DUPONT[4], Susana JORQUERA[3], Pauline MARTINET[2], Jean-Francois
RIBAUD[5], and Felipe TOLEDO[3]
1) Institut Pierre Simon Laplace, CNRS, Ecole Polytechnique de Paris, Institut Polytechnique de Paris, Paris, France
2) Meteo-France, Toulouse, France
3) Laboratoire Atmosphères, Milieux, Observations Spatiales/UVSQ/CNRS/UPMC, 78280 Guyancourt, France
4) Institut Pierre-Simon Laplace, École Polytechnique, UVSQ, Université Paris-Saclay, 91128 Palaiseau, France
5) Laboratoire de Météorologie Dynamique, École Polytechnique, 91128 Palaiseau, France.
* corresponding author: Cheikh DIONE, cdione@ipsl.fr
**Abstract:**
In this study, we use a synergy of in-situ and remote sensing measurements collected during the Southwest
FOGs 3D experiment for processes study (SOFOG3D) field campaign in autumn 2019 and winter 2020, to
analyze the thermodynamic and turbulence processes related to fog formation, evolution, and dissipation
across southwestern France. Based on a unique dataset with a very high resolution and a fog conceptual
model, an analysis of the four heaviest fog episodes (two radiation fogs and two advection-radiation fogs) is
conducted. The results show that radiation and advection-radiation fogs form under deep and thin
temperature inversion, respectively. For both fog categories, the transition period from stable to adiabatic fog
and the fog adiabatic phase are driven by vertical mixing associated with an increase in turbulence in the fog
layer due to mechanical production (turbulence kinetic energy (TKE) up to 0.4 $m^2\,s^{-2}$ and vertical velocity
variance ($\sigma_w^2$) up to 0.04 $m^2\,s^{-2}$) generated by brisk wind at the supersite (advection). The dissipation time is
observed at night for the advection-radiation fog case studies and during the day for the radiation fog case
studies. Night-time dissipation is driven by horizontal advection generating mechanical turbulence (TKE at
least 0.3 $m^2\,s^{-2}$ and $\sigma_w^2$ larger than 0.04 $m^2\,s^{-2}$). Daytime dissipation is linked to the combination of thermal
and mechanical turbulence related respectively to solar heating (near surface sensible heat flux larger than 10
$W\,m^{-2}$) and advection. Through a deficit of the fog reservoir of liquid water path, the fog conceptual model
estimates the dissipation time at least one hour before the observed dissipation for radiation fog cases. It
gives a better estimate of the fog dissipation time for advection-radiation cases. This study also demonstrates
the importance of using instrumental synergy (with microwave radiometer, wind lidar, weather station, and
cloud radar) and a fog conceptual model to better predict fog characteristics and dissipation time at
nowcasting ranges.
*Key words:* Fog conceptual model, radiation/advection fog, fog life cycle, turbulence, Southwestern France,
SOFOG3D



**1. Introduction**

Fog is an extreme meteorological phenomenon forming in several regions of earth under different atmospheric conditions depending on the season and location (Gultepe et al., 2007). It is defined by the suspension of water droplets in the lowest troposphere which reduces the horizontal visibility to lower or at least 1000 m. Fog has significant negative impacts on air, road and marine traffic causing large economical and human losses (Bartok et al., 2012, Bartoková et al., 2015, Huang and Chen, 2016). It also has a high impact on solar energy, particularly in the mid-latitudes during Autumn and Winter. Based on in-situ measurements, several studies have focused on fog formation at different regions and highlighted the main processes leading to its initiation allowing to define four categories of fog: radiation fog (Price 2019), advection-radiation fog (Gultepe et al., 2007, 2009; Niu et al., 2010a, b, Dupont et al., 2012), advection fog (Koračin et al., 2014; Liu et al., 2016, Fernando et al., 2021), and precipitation fog (Tardif and Rasmussen, 2007; Liu et al., 2012). According to the literature, several processes are identified to drive fog evolution and dissipation depending on each category. Fog formation requires low intensity of turbulence (Nakanishi 2000; Bergot 2013; Price 2019)

Dhangar et al., 2021 found that optically thin fog develops under low-turbulence kinetic energy and the transition to dense fog is observed when the turbulence increases and reaches enough values to allow the vertical mixing of the fog layer. The dissipation of radiation fog is usually observed after sunrise and linked with the increase in solar heating leading to the evaporation of water drops and a vertical mixing of water vapor (Roach, 1995; Haeffelin et al., 2010; Maalick et al., 2016). Bergot et al., 2015 relied on large eddy simulations (LES) to characterize the role of dry downdrafts in allowing solar radiation to reach the ground and increasing the turbulence. Additionally, Pauli et al., 2022 studied the climatology of fog and low stratus cloud formation and dissipation times in Central Europe using satellite data and showed that fog dissipation is also often related to topography. The dissipation processes are more difficult to study then the fog formation processes, due to the complexity of fog's scale. At the state of the art, based on case studies, numerical weather prediction models (Philip et al., 2016, Bell et al., 2022) and high resolution models (Price et al., 2018, Ducongé et al., 2020, Fathalli et al., 2022) up to LES (Bergot et al., 2015, Mazoyer et al., 2017) have the ability to simulate fog formation in several complex areas. However, they have difficulties in simulating the processes driving fog evolution over land in real time (Steeneveld *et al*., 2015, Price et al., 2015, Román-Cascón et al., 2016; Wærsted et al., 2019; Pithani et al., 2020, Boutle *et al*., 2022).



Toledo et al., 2021 developed a one-column conceptual model of adiabatic continental fog
allowing to define key fog metrics as the equivalent fog adiabaticity by closure and the reservoir of
liquid water path (RLWP) that can be estimated in real-time and allowing a diagnostic of fog
evolution. Based on seven years of measurements collected at SIRTA (Site Instrumental de
Recherche par Télédétection Atmosphérique), a French observatory located at Palaiseau/France,
Toledo et al., 2021 have validated their model on the timing of fog dissipation based on the RLWP.
The limitation of this model is that the estimation of the reservoir depends on fog specific
parameters and does not take into account local (turbulence) or large scale processes (advection).
Indeed, to further understand uncertainties associated with the estimation of the RLWP, the
validation of the model using data from other measurement sites having a large occurrence of fog is
another step before using it in nowcasting ranges.
Understanding the life cycle of fog is an imperative for numerical weather prediction models
in order to set up an effective and efficient early warning system to reduce the socio-economic
impacts of this phenomenon in areas with high occurrence of fog. Thus, finding the right
instruments on which this warning system will be based is also another challenge that can be partly
resolved by field campaigns combining both in-situ and remote sensing measurements and
numerical simulations. At the state of the art, nowcasting fog requires more efforts in in-situ
measurements and modeling. In this context, the SOuth westFOGs 3D (SOFOG3D) project, led by
Météo-France, was designed to document local processes involved in fog formation, evolution and
dissipation to better improve its predictability in numerical weather prediction models in the
Southwestern France.
In order to improve our understanding of the processes driving the fog life cycle and to validate the
fog conceptual model from Toledo et al., 2021 on another region than the one on which it has been
developed, the current study aims at identifying the main dynamical and thermodynamic processes
driving fog's formation, evolution, and dissipation in the framework of SOFOG3D project. Using
an instrumental synergy of in-situ and remote sensing measurements and the fog conceptual model,
the phenomenology of fog and the different phases driving its evolution are deeply analyzed
considering four heavy fog case studies observed over Southern France during Winter 2019-2020.
This paper is structured into five sections. The datasets and methodological approach are
described in the following section. Section 3 gives an analysis of the processes involved in fog
evolution based on two different categories of fog formation phenomenology. Section 4 of this



manuscript includes a discussion on the thermodynamical and turbulent processes driving the fog
phases and Section 5 presents the conclusion.
**2. Data and methodology**

In a mesoscale context, the SOFOG3D field experiment is located in Southwestern France,

in the Aquitaine region (Fig. 1a). The field campaign was carried out during the Autumn 2019 and
Winter 2020 period leading to 15 intensive observation periods (IOPs). A unique dataset has been
collected across a complex region with a very contrasted topography. This region is bordered in the
east by the "Massif Central", in the west by the Atlantic Ocean, in the north by Bordeaux and in the
south by the "Pyrenees". In the region, several dynamical effects can be observed such as sea
breeze, land breeze, and mesoscale foehn circulations influencing the fog life cycle. At the local
scale, the supersite under focused here is bordered by two rivers: "La Garonne" to the East and
"L'Eyre" to the west (Fig. 1a). These two rivers and the surrounding surface heterogeneities can
modulate the fog formation and dissipation times. During the campaign, several in-situ and remote
sensing measurements were jointly deployed in the studied area of SOFOG3D. In this paper, our
analysis focuses on the data collected in the surroundings of the supersite at Charbonnière, the most
instrumented site (Fig. 1b) during the field campaign. Below, the descriptions of the in-situ and
remote sensing measurements and then the fog conceptual model are presented with emphasis on
the main meteorological variables used in the study.
**2.1 Dataset**
**2.1.1 Surface measurement data**

A network of surface weather stations was installed in the study domain of SOFOG3D at the

vicinity of Charbonnière, to document the spatial variability of fog and surface heterogeneities at
the local scale (Fig. 1b). Four weather stations were also deployed around the supersite in a
northeast-southwest transect (Fig. 1b). These stations were installed at Moustey, Cape Sud, Tuzan
and Noaillan, almost at the same altitude, and operated continuously with very high temporal
resolution (0.1 s time interval) during the period from 18 October 2019 to 31 December 2020. In
addition to temperature, pressure, relative humidity sensors and anemometer, a scatterometer
provided the visibility used to estimate fog formation and dissipation times at each station.
Temperature data are used to characterize the spatial variability of the radiative cooling. Wind



speed and direction are used to get an indication of the local circulations and their association with
air mass advection (spatial coherence of wind) and source of turbulence.
In this study, fog occurrence is defined using the visibility at the supersite based on an
algorithm developed by Tardif and Rasmussen, 2007. This algorithm consists of dividing visibility
time series into 10 min blocks. A fog block means that half of the visibility measurements during a
10 min period are below 1000 m. Blocks are characterized by a positive or negative construct. A
positive construct indicates that five consecutive blocks of which the central block is fog and at least
two other blocks are also fog blocks. The opposite means a negative construct. Thus, the fog
formation time corresponds to the first fog block in the first positive construct encountered. The fog
dissipation time corresponds to the last fog block in the last positive construct before either a
negative construct or three consecutive non-fog blocks are encountered. This algorithm discards fog
events shorter than 1 hour.
Meteo-France installed in a fallow field near the supersite, several sensors as Licor analyzers
and sonic anemometers to continuously measure the near-surface (3 m a.g.l) meteorological
conditions (air temperature and relative humidity) and pressure at 0.3 m a.g.l) and the three
components of the wind at 10 m a.g.l. These instruments provided high frequency data at 20 Hz. In
this study, to document fog dissipation processes, we use sensible heat flux (SHF), turbulence
kinetic energy (TKE), and vertical velocity variance ($\sigma_w^2$). These variables are estimated using the
Eddy-covariance methods (Foken et al., 2004, Mauder et al., 2013) calculated every 30 minutes
after a high quality control of the data. More details on the data can be found in Canut, 2020.
**2.1.2 Observation of cloud characteristics**
For the monitoring of cloud layers, a BASTA cloud radar (Delanoë et al., 2016) was
deployed at Charbonnière and a CL51 Ceilometer at Tuzan (7.4 km northwest of Charbonnière)
(Fig. 1b).
BASTA is a 95-GHz cloud radar manufactured by the Laboratoire Atmosphères, Milieux,
Observations Spatiales (LATMOS) with an absolute calibration method for frequency-modulated
continuous wave (FMCW) cloud radars based on corner reflectors (Toledo et al., 2020). From 7
November 2019 to 12 March 2020, the radar was operated continuously with a vertical pointing
mode having three vertical resolutions (12.5 m, 25 m, and 100 m). It provided radar reflectivity and
Doppler velocity. The lowest mode, having its first available gate at 37.5 m a.g.l and 12.5 m of
vertical resolution, is used to estimate the cloud top height (CTH) which gives the fog thickness at a





time resolution of 30 seconds. It also provides the level of highest concentration of droplets in the
fog layer. The CTH is estimated using a radar reflectivity threshold of -34 dBZ.
The CL51 is manufactured by Vaisala and automatically provided three estimates of cloud
base height (CBH) allowing the detection of cloud decks every 30 seconds with a vertical resolution
of 15 m. from 10 October 2019 to 2 April 2020. In this study, we use the lowest CBH, which
corresponds to the base height of stratus cloud lowering or lifting when fog forms or dissipates,
respectively. More information on the data provided by the CL51 can be found in Burnet, 2021.
**2.1.3 Temperature and wind profiling**
A microwave radiometer Hatpro (MWR) manufactured by Radiometer Physics GmbH
(RPG) was installed at the supersite to characterize thermodynamic atmospheric conditions during
the field campaign. From 4 December 2019 to 9 May 2020, the MWR operated continuously at the
supersite using two spectral-bands: the K-band which 22.24-31 GHz used for the retrieval of
humidity profiles, integrated water vapor (IWV) content and liquid water path (LWP), and the V-
band which 51-58 GHz to retrieve temperature profiles. In order to improve the vertical resolution
in the boundary layer, the MWR was set up to scan in 10 elevation angles every 10 minutes with a
zenith pointing each 1 second. Using neural networks, brightness temperatures measured by the
MWR are inverted to temperature and humidity variables. More details on this method can be found
in Martinet et al., 2022. Comparing temperature and humidity profiles retrieved by the MWR with
radiosonde data, Martinet et al., 2022 found that air temperature has cold biases below 0.5 K in
absolute value below 2 km but increases up to 1.5 K above 4 km altitude. The low biases in the
lowest atmosphere allow a good estimation of the lowest temperature inversion under focus in this
study. For each case study, the transition from stable to adiabatic fog is estimated using the static
atmospheric stability in the lowest atmosphere computed using the temperature profile. The air
temperature profiles are also used to characterize the atmospheric conditions linked to the
development of fog at Charbonnière. For the absolute humidity, the maximum dry bias of the MWR
is around 1.4 g m$^{-3}$ in the lowest troposphere up to 1.7 km and becomes wet above (0.3 g m$^{-3}$). The
small biases in humidity profiles shows that the LWP accuracy is in the scope of those defined in
the literature (Crewell and Löhnert, 2003; Marke et al., 2016). The LWP is a key parameter to
consider for the microphysical characteristics of fog and is used in the conceptual model. More
information regarding the data can be found in Martinet, 2021.



The WindCube lidar becomes a common instrument used in documenting very low
atmospheric phenomena such as turbulence (Liao et al., 2020; Kumer et al., 2016). Dias Neto et al.,
2023 demonstrated the usefulness of the wind speed and direction estimated using the WindCube
V2. Comparing wind from WindCube V2 with GPS radiosonde, they found low biases of 0.52 m s$^{-1}$
and 0.37° for the wind speed and direction, respectively. To investigate the dynamics of the
atmosphere at the supersite, a WindCube V2 lidar manufactured by Leosphere was deployed by
Meteo-France during the field campaign to provide from 1 October 2019 to 10 April 2020, the wind
measurements at 10 levels ranging from 40 m to 220 m above ground level (a.g.l). The
measurements made at a 1 Hz frequency and a 20 m vertical resolution provided the estimation of
turbulence parameters such as the turbulent kinetic energy (TKE). The TKE is computed every 30
minutes using the horizontal wind component at the high resolution. It is used in this study to
analyze the role of turbulence within the foggy-layer to further characterize fog formation,
evolution, and dissipation. More details on the WindCube lidar data can be found in Canut et al.,

2022.

### 2.1.4 Fog adiabaticity and reservoir

To further understand fog characteristics, it is essential to focus our analysis on several
variables related to the formation, evolution and dissipation of fog. Fog adiabaticity and reservoir
are key metrics driving the life cycle of fog. They are estimated using the fog conceptual model
(Toledo et al., 2021) developed at SIRTA. This model is a uni-dimensional model inspired by
previous numerical models for stratus clouds (Betts, 1982, Albrecht et al., 1990; and Cermak and
Bendix, 2011). The basic hypothesis is to consider a well-mixed fog layer and to express the
increase in height of the fog liquid water content as a function of the local adiabaticity and the
negative of the change in the saturation mixing ratio with height ($\Gamma_{ad}(T,P)$) (equation A1). Fog
liquid water path is parameterized as a function depending on the equivalent fog adiabaticity ($\alpha_{eq}$)
and the CTH (equation A3). The equivalent fog adiabaticity is used to characterize the buoyancy in
low clouds. $\alpha_{eq}$ varies depending on the in-cloud mixing parameter $\beta$ and is expressed as $\alpha_{eq} = (1-\beta)$
(Betts, 1982 and Cermak and Bendix, 2011). For low-level clouds, as stratus and stratocumulus, $\alpha_{eq}$
is between 0.6 and 0.9 (Braun et al., 2018) indicating sufficient buoyancy in the cloud layer with an
adiabatic profile. To parameterize this parameter in the fog conceptual model, Toledo et al., 2021
used an inversion of Eq. (A3) to define a fog adiabaticity from closure ($\alpha_{eq}^{closure}$) given in equation
(1). $\alpha_{eq}^{closure}$ depends on the accumulated liquid water content (LWC) at the fog base (LWC$_o$), fog



thickness (e.g. CTH), the LWP and the adiabaticity. The adiabaticity lapse rate is a function of air
temperature and pressure. Toledo et al., 2021 found that the equivalent fog adiabaticity from closure
is negative when the LWP is below 30 g m$^{-2}$. They defined the transition phase from stable to
adiabatic conditions when the equivalent fog adiabaticity from closure is around 0.5. In the
conceptual model, this parameter is estimated only for a CTH below 462.5 m with free cloud above.
$$\alpha_{eq}^{closure} = \frac{2\left(LWP - LWC_0 CTH\right)}{\Gamma_{ad}\left(T,P\right)CTH^2}$$    (1)

Considering that adiabatic fog exists because the liquid water path in its thickness is strictly

greater or equal to its critical liquid water path (CLWP) (Toledo et al., 2021), it is possible to define
an associated quantity named the fog reservoir of liquid water path (RLWP). The RLWP is defined
as the difference between fog current liquid water path and the critical value, as shown in equation
2. It depends on the critical liquid water content (LWCc) (A.4), the adiabaticity and fog thickness.
The calculation of fog RLWP can be used to anticipate the dissipation or thickening of the fog in
the coming minutes or hours. Based on 20 fog cases at SIRTA, Toledo, 2021 found that for a
RLWP > 30 g m$^{-2}$ in a given time instant, fog does not dissipate within the following 30 minutes. He
also showed that the RLWP trend decreases before fog dissipation time and increases when fog is
persisting. This behavior motivates the analysis of the RLWP trend in this study to improve the
characterization of the different fog phases.
$$RLWP = LWP - CLWP = LWP - \frac{1}{2}\alpha_{eq}\Gamma_{ad}\left(T,P\right)CTH^2 - LWC_c CTH$$    (2)

The number of fog events observed during the SOFOG3D field campaign is not sufficient to

calibrate the fog conceptual model in southeastern France as in SIRTA (Toledo et al., 2021). In this
study, we use the model with its parametrization at SIRTA to further characterize the different
phases observed in the lifetime of fog based on single identified case studies. The model is
performed when the visibility is lower than 1000 m. $\alpha_{eq}^{\ closure}$ is used to characterize the fog transition
from stable phase to adiabatic phase. The RLWP gives information about the predictability of fog
dissipation time at nowcasting range. More details on the fog conceptual model is given in
appendices and can be found in Toledo, 2021.
**2.2 Case studies and methodological approach**

For the whole SOFOG3D campaign, based on the fog defined criteria described in section

2.2.1, 31 fog events are identified during 31 October 2019 - 26 March 2020 period. For each one, a





visual expectation of the time-height cross-section of the radar reflectivity from BASTA cloud radar and the cloud base height from the Ceilometer was carried out. We selected the four most developed fog episodes, namely case studies 1 (IOP 5), 2 (IOP 6), 3 (IOP 11) and 4 (IOP 14).

As in Toledo et al., 2021 (their Fig. 3), Figure 2 shows the equivalent adiabaticity by closure versus LWP and CTH for the 4 fog case studied. It indicates that $\alpha_{eq}^{closure}$ researches 0.5 when LWP > 20 g m$^{-2}$ and the CTH > 150 m which should be the conditions favorable for the fog to become optically opaque to the infrared radiation. At the supersite, the LWP observed during that transition is lower than the threshold at SIRTA (LWP > 30 g m$^{-2}$) (Wærsted et al., 2017 and Toledo et al., 2021). However, there is a consistency between both sites on the computation of the equivalent adiabaticity by closure. This legitimises the choice of the four days, and motivates the use of the $\alpha_{eq}^{closure}$ in this study to define the transition phase between stable and adiabatic fog.

For the selected case studies, Table 1 contains the fog formation and dissipation times, fog formation types, and fog duration at the supersite. For all selected fog events, the formation time of fog is observed between 20:40 and 22:40 UTC and the dissipation time varies from night to daytime. These selected fogs triggered by radiation (2 cases) or advection-radiation (2 cases) processes.

For each selected case study, temperature profiles from the MWR, radar reflectivity profiles from the BASTA cloud radar and the equivalent fog adiabaticity derived from the conceptual model are used to define the four fog phases characterizing the fog evolution: fog pre-onset, stable fog, adiabatic fog, and fog dissipation. Note that an important time of the fog life cycle is the transition time between stable and adiabatic fog. Each fog phase is defined as following:

1/ Fog pre-onset is defined as the two hours preceding fog onset associated with cloud free conditions.

2/ In the four cases studies, the stable phase starts at fog onset. It is characterized by a stable temperature profile in the lowest 100 m of the atmosphere.

3/ The transition time separating the stable and adiabatic phases can be defined differently depending on the meteorological variables considered. Price et al., 2011 defined this transition time as the time when the air temperature is constant in the fog lowest layer (1.5 - 50 m a.g.l). Toledo et al., 2021 found that the transition is observed when the equivalent fog adiabaticity by closure is increasing between 0 and 0.5. In this study, for a better definition of this period, we take into account the static stability given by the hourly profiles of mean air temperature from the MWR, the fog geometry (CTH) from the cloud radar, and the $\alpha_{eq}^{closure}$ from the conceptual model. Indeed, the





transition period is defined as the time when the temperature profile becomes unstable or neutral in
the 0-75 m a.g.l layer, while the fog CTH increases with time, and $\alpha_{eq}^{closure}$ increases from 0 to about
0.5. Note that the thickening of the fog is associated with the elevation of the level of the maximum
radar reflectivity. The transition phase starts when $\alpha_{eq}^{closure} < 0.5$, the CTH suddenly increases more
than 25 m in 5 minutes under a stable or neutral layer. This phase ends when $\alpha_{eq}^{closure}$ reaches 0.5 and
the fog layer becomes neutral or unstable.
4/ Fog adiabatic phase is characterized by $\alpha_{eq}^{closure}$ around 0.5, a neutral or unstable
temperature profile, and a radar reflectivity that increases with increasing altitude and peaks a few
tenths of meters below cloud top.
5/ Fog dissipation phase is defined as being the period between 30 minutes before and after
dissipation time (when horizontal visibility becomes greater than 1 km). Since the fog dissipation
time does not appear abruptly, as it is driven by thermodynamical processes, we consider this time
range to further document them.
Based on these fog phase definitions, in the following, we describe the four case studies. For
each fog event, we document, using the fog conceptual model and the instrumental synergy, the
processes involved in the evolution of fog in each of these phases, in order to identify the main
processes driving the fog life cycle.
***Table 1 :*** *Case study number, fog onsets, type of fog formation, fog dissipation times, fog duration*
*and type of fog dissipation for the four documented case studies. Time is in UTC. Dates are in the*
*format "dd/mm/yyyy". "dd" indicates the day, "mm" the month and "yyyy" the year.*

| Case study number | Formation time | | Fog types | Dissipation time | | Fog duration (hh:min) |
| --- | --- | --- | --- | --- | --- | --- |
| | Date dd/mm/yyyy | Hours (UTC) | | Date dd/mm/yyyy | Hours (UTC) | |
| 1 | 28/12/2019 | 22:40 | Radiation | 29/12/2019 | 11:00 | 12:20 |
| 2 | 05/01/2020 | 20:40 | Radiation | 06/01/2020 | 08:40 | 12:00 |
| 3 | 08/02/2020 | 20:40 | Advection-radiation | 09/02/2020 | 03:40 | 7:00 |
| 4 | 07/03/2020 | 21:20 | Advection-radiation | 08/03/2020 | 04:00 | 6:40 |

**3. Fog formation, evolution, and dissipation processes**
**3.1 Case study 1 (IOP 5) analysis**



Figures 3a and 3b indicate the time-cross sections of the radar reflectivity estimated from
BASTA cloud radar during case study 1, on the 28-29 December 2019, respectively up to 600 m
and 12000 m. They show a clear sky before fog formation time at 22:40 UTC on 28 December
2019. During fog evolution, cloud free conditions are observed above the fog top height until 09:00
UTC when sparse thin high-altitude clouds occur above the cloud radar. Figure 3c presents a quasi-
homogeneous fog formation time between the three sites and heterogeneous dissipation time. At
Charbonnière, fog dissipated at 11:00 UTC, on 29 December 2019 and two hours earlier at
Noaillan. At all sites, low temperatures below 4 °C (Fig. 3e) are observed during the fog period.
Near the surface, light wind (< 1 m s$^{-1}$) are recorded at all sites from fog pre-onset to fog
stable/adiabatic transition times (Fig. 3d and 3f).
The fog pre-onset is marked by a double stratification of the atmospheric boundary layer
with a thin inversion from surface up to 100 m and deep and strong inversion (14 °C km$^{-1}$) above
(Fig. 4a). Atmospheric conditions are dominated by an easterly wind that reaches 5 m s$^{-1}$ above 100
m a.g.l which could be considered as a nocturnal low-level jet (Fig. 4d). The mean cooling rate near
the surface is -0.9 °C h$^{-1}$. The strong decrease in temperature is associated with surface radiative
cooling (cloud free), negative SHF (-0.23 W m$^{-2}$) (Fig. 4h), near surface low wind (0.61 m s$^{-1}$) (Fig.
3d and 3f) and very low thermal turbulence (TKE = 0.18 m$^2$ s$^{-2}$ and $\sigma_w^2$ = 0.002 m$^2$ s$^{-2}$). These
conditions lead to thermally-stable atmospheric conditions which are favorable for fog formation
(Table 1). The fog onset slightly precedes the minimum of SHF.
The fog stable phase lasts around 6 h (22:50 - 05:00 UTC). Near the surface, it is
characterized on average by a very low radiative cooling rate (-0.18 °C h$^{-1}$), an almost zero SHF , an
easterly light wind (0.78 m s$^{-1}$), low turbulence (TKE = 0.12 m$^2$ s$^{-2}$, $\sigma_w^2$ = 0.01 m$^2$ s$^{-2}$), and a negative
$\alpha_{eq}^{closure}$ (-1.3) (Fig. 4e), a low LWP of 2.18 g m$^{-2}$ (Fig. 4g), a slight increase in time of the fog
thickness up to 50 m, and a relatively stable temperature inversion height. During this phase,
turbulence, LWP and RLWP are sufficiently low to maintain thermally-stable fog with an horizontal
visibility of 736 m on average.
For this case, the transition time from stable fog to adiabatic fog is observed between 05:00
and 07:00 UTC at the supersite. It corresponds to the lowest visibility (198 m) and is illustrated by a
transition in the vertical profiles of air temperature (Fig. 4a) from stable at 05:00 to unstable at
06:00 UTC. The transition is materialized by a deepening of the cold layer. At 05:00 UTC the
coldest temperature is at the surface. At 06:00 UTC, the minimum temperature is observed at 50 m
a.g.l. At that time, the vertical profile of radar reflectivity increases with height, indicating a vertical



development of fog (Fig. 4b). At the end of this phase, $\alpha_{eq}^{closure}$ reaches 0.5 which is consistent with
the threshold obtained at the SIRTA site by Toledo et al., 2021. The mean SHF reaches 4.4 W m$^{-2}$
and around 10 W m$^{-2}$ at the phase end (Fig. 4h). The wind speed at 10 m a.g.l increases to 1.14 m s$^{-1}$
and shifts in direction from East to Southeast. The TKE remains constant and the $\sigma_w^2$ significantly
increases to 0.01 m$^2$ s$^{-2}$. Vertical velocity variance values observed are higher than the threshold
fixed by Price et al., 2019 for a thermally-stable surface layer. This increase in turbulence indicates
a vertical mixing in the fog layer. The LWP and RLWP peak at the end of the transition phase
consistently with a decrease in visibility. Due to the simultaneous increase in SHF, TKE and $\sigma_w^2$,
the transition phase is driven by both thermal and mechanical turbulence.

The fog adiabatic phase is observed between 07:00 and 11:00 UTC (4 h duration) at the

supersite. This phase is characterized by a vertical development of fog up to 185 m (Fig. 4b) and the
arrival of sparse high clouds (Fig. 3a and 3b) associated with the lowering of the temperature
inversion top height above the fog top (Fig. 4c). Note that these clouds have no effect on the
radiative cooling at the top height of the fog. The fog layer becomes warmer (+0.77 °C h$^{-1}$ on
average) and its LWP and RLWP reach 26.16 g m$^{-2}$ and +6.38 g m$^{-2}$, respectively. The turbulence
gradually increases in the fog layer (Fig. 4f) (TKE = 0.28 m$^2$ s$^{-2}$) due to an increase of the horizontal
wind speed (2.4 m s$^{-1}$) and its shift from southeasterly to easterly (Fig. 4d). In the same way, the
vertical velocity variance increases to 0.04 m$^2$ s$^{-2}$ and is driven by the vertical wind shear and the
increase in SHF (12.9 W m$^{-2}$) (Fig. 4h). For this case study, the moderate mechanical and thermal
turbulence causes vertical mixing in the fog layer, which slightly increases the surface horizontal
visibility (370 m).

At the supersite, the fog dissipates after sunrise under cloud free atmosphere above its top

height. The SHF continues to increase (Fig. 4h) due to solar radiation. During this phase, the RLWP
becomes negative (-11.39 g m$^{-2}$) when the CTH increases significantly, in spite of the increase of
the LWP (maximum of 43.34 g m$^{-2}$), while $\alpha_{eq}^{closure}$ remains around 0.63. Based on the RLWP, the
fog conceptual model would predict a deficit of liquid water in the fog layer one hour before the
lifting of its base height (Fig. 4g). The fog dissipation phase is induced by the increase of the
vertical mixing generated by the thermal and mechanical turbulence associated with TKE values
larger than 0.4 m$^2$ s$^{-2}$ (Fig. 4f). The fog dissipation phase is marked by the daytime atmospheric
convection associated with significant SHF (22.02 W m$^{-2}$) generating thermal turbulence ($\sigma_w^2$ = 0.06
m$^2$ s$^{-2}$), which allows more vertical mixing and warming of the daytime atmospheric boundary layer.



In summary, for this fog event, the fog conceptual model is consistent with the in-situ measurements of turbulence on the timing of the different fog phases. It has provided additional elements for understanding the different phases of the fog life cycle.

**3.2 Case study 2 (IOP 6) analysis**

Radar reflectivity time-cross sections derived from BASTA cloud radar during case study 2 (IOP 6) on the 5-6 January 2019 indicate that clear weather precedes fog formation at 20:40 UTC on 5 January 2020 (Fig. 5a and 5b). Fog develops below the dry, warm and cloud free stable atmospheric boundary layer (Fig. 5c). This case presents a spatial variability of fog formation time. The fog lasts 12 h and completely dissipates around 08:40 UTC, on 6 January 2020 at the supersite (see Table 1), while it dissipates earlier at Noaillan at 04:30 UTC. At all sites in the studied area, cold atmospheric conditions prevailed during the whole episode (Fig. 5e). The surface wind speed is moderate (< 3 m s$^{-1}$) and quite homogeneous in the studied area (Fig. 5d and 5f). The wind direction changed several times during the fog's evolution.

As in case study 1, before fog formation, hourly vertical profiles of temperature from the MWR (Fig. 6a) indicate a double stratification of the low atmosphere under an easterly low-level jet (Fig. 6d). Near surface air temperature is negative (Fig. 5e) and indicates frozen surface. These conditions are associated with an anticyclonic system across central Europe (not shown). During the fog pre-onset phase, the mean cooling rate at the supersite is -0.7 °C h$^{-1}$ (Table 2). The continued decrease in temperature combined with the negative surface SHF (-0.17 W m$^{-2}$), southerly very low wind (0.2 m s$^{-1}$) at near surface, very low vertical velocity variance ($\sigma_w^2 < 0.003$ m$^2$ s$^{-2}$) and low TKE (0.06 m$^2$ s$^{-2}$) reveal that atmospheric conditions favorable to fog formation are driven by surface radiative cooling (Table 1), leading to a thermally-stable surface layer as in case study 1. Again, the fog onset precedes by a few minutes the minimum of SHF.

The fog stable phase is observed from 20:40 UTC to 03:00 UTC (3 h 20 min duration) under cloud-free conditions above fog top height. It is characterized by a thin fog (71 m) under a very deep temperature inversion (Fig. 6c), and light varying wind (Fig. 6d). Negative values of the equivalent fog adiabaticity by closure (-0.69) associated with decrease in temperature (-0.13 °C h$^{-1}$) (Fig. 5e), very low mean LWP (1.66 g m$^{-2}$) (Fig. 6g), and low turbulence (TKE = 0.09 m$^2$ s$^{-2}$ and $\sigma_w^2$ = 0.009 m$^2$s$^{-2}$) are sufficient conditions to maintain a thermally stable, optically-thin fog (242 m of horizontal visibility), as in case study 1. The continued increase of TKE in the fog layer (Fig. 6f), and surface SHF (Fig. 6h) triggered the start of the transition phase, limiting the duration of the stable phase compared to case 1 (IOP 5).





For case study 2, the fog transition phase is observed from 00:00 UTC to 02:00 UTC (2 h of
duration) at the supersite (Fig. 6a and 6b). Its characteristics are similar to those found in case study
1 but the LWP (7.18 g m$^{-2}$), RLWP (+3.55 g m$^{-2}$), cooling rate (-0.007 °C h$^{-1}$) are lower and the
TKE (0.23 m$^2$.s$^{-1}$) and SHF (7.76 W m$^{-2}$) larger. As in case study 1, these turbulent conditions allow
a vertical mixing of the fog layer indicating its transition towards adiabatic fog.
Fog adiabatic phase is observed from 02:00 UTC to 08:40 UTC at the supersite. The first
period from 02:00 UTC to 05:00 UTC is marked by a $\alpha_{eq}^{closure}$ larger than 0.5 and a strong increase in
temperature (+2 °C), LWP (42 g m$^{-2}$), and a positive RLWP until 04:30 UTC. The temperature
inversion above the fog layer strengthened and its top height lowered. The TKE in the fog layer and
the vertical velocity variance continue to increase (TKE > 0.2 m$^2$ s$^{-2}$ and $\sigma_w^2$ > 0.02 m$^2$ s$^{-2}$). The SHF
oscillates around 10 W m$^{-1}$. These conditions are favorable for the deepening of the fog by vertical
mixing (see Fig. 5a and 6b). The second period from 05:00 UTC to 08:40 UTC is characterized by
the $\alpha_{eq}^{closure}$ lower than 0.5, a decrease in surface temperature, stable base and top height of the
temperature inversion, a sharp decrease in LWP, fog top height and RLWP (oscillating around 0 g
m$^{-2}$), while the horizontal visibility increases and then decreases again. The decrease in turbulence
(TKE < 0.2 m$^2$.s$^{-2}$) is linked to the decrease in wind speed in the fog layer, while the vertical
velocity variance remains significant ( $\sigma_w^2$ > 0.02 m$^2$ s$^{-2}$) with positive SHF. During the second half
of the adiabatic phase, the fog layer that contains less than 20 g m$^{-2}$ liquid water is not very resilient
to the significant turbulence, as shown by the very low RLWP values and rapidly changing
horizontal visibility.
The decrease in LWP seems to be driven by a possible phase change (water droplet to snow
droplets) of the water droplets inducing a cooling in the fog layer and an increase in horizontal
visibility. The dissipation of the mechanical turbulence favors the lowering of the fog thickness.
These processes seem to be linked to the formation of snowflakes in the fog layer with fall due to
their gravity which is consistent with the visual observations of scientists operating at the supersite,
who reported frost on the tethered balloon.
As in case study 1, at the supersite, fog dissipates in the morning at 08:40 UTC, around
sunrise. The RLWP predicted the fog dissipation at 07:30 UTC, one hour fifteen minutes before its
total dissipation time. The surface vertical velocity variance became larger than 0.04 m$^2$ s$^{-2}$ and the
TKE in the fog layer higher than 0.4 m$^2$ s$^{-2}$, the $\alpha_{eq}^{closure}$ oscillated around 0.5. These atmospheric
characteristics in the fog layer are linked to the increase in turbulence associated with the increase
of the wind speed (Fig. 5d) and the SHF (Fig. 5h), both induced by the convective mixing due to





solar radiation. Therefore, as in case study 1, the dissipation of fog is driven by the turbulence
associated with mechanical and thermal processes.

### 3.3 Case study 3 (IOP 11) analysis

Radar reflectivity cross-sections on the 8-9 February 2020 (IOP 11) (Fig. 7a and 7b) indicate
that this fog event is characterized by an early formation of fog at 20:40 UTC. Fog formation is
preceded by a short rain (8.36 mm at Moustey) period produced by a stratocumulus cloud (Fig. 7b).
After the rain, the water vapor in the lowest atmosphere starts to condensate as an ultra-low stratus
cloud due to radiative cooling. From fog formation time up to 03:00 UTC,the sky is clear above the
fog at the supersite. These atmospheric conditions allow a radiative cooling which favors the
stabilization of the surface layer. Figure 7c indicates a spatio-temporal variability of fog formation
time during a period of strong decrease in near surface temperature (Fig. 7e) at the beginning of the
night and relatively light westerly wind (Fig. 7f and 7d). The formation of the fog started from the
West and spread toward the East, illustrating a West-East gradient of fog formation in line with the
westerly wind blowing in the studied area. During fog evolution, there is a spatial heterogeneity of
temperatures up to 4°C between Moustey (western and coldest site) and Noaillan (eastern and
warmest site). On the other hand, the dissipation is fairly homogeneous at all the sites, consecutive
to an increase in air temperature and wind speed, and a shift in the wind direction (south-south-east
to south), except at Noaillan where fog dissipation occurs earlier as visibility and temperature are
higher than at the other sites. The fog dissipates at 03:40 UTC when the low atmosphere becomes
neutral or unstable and the maximum radar reflectivity decreases and jumps in height (Fig. 8b). Just
after the fog dissipation time, high clouds appear around 10 000 m height, characterising the change
in air mass by advection.
Figure 8c shows that for this case study, the temperature inversion forms after the formation
of the ultra-low stratus associated with the advection of the westerly Atlantic flow near the ground
(Fig. 8d). The westerly flow brings wet and mild air over land and contributes to reduce the surface
radiative cooling. The temperature inversion formed at the same time as the base of the stratus
touches the ground, justifying the classification of fog formation by advection-radiation processes
(Ryznar, 1977).
The formation of the fog considerably modified the dynamics of the low-level atmosphere
by slowing down the radiative cooling, thus creating a thin layer of temperature inversion around
250 m thick with a low intensity of about 3 °C. The fog stable phase is observed from 20:40 UTC



and 23:00 UTC at the supersite. It is characterized by a clear sky above the fog top, a decrease in
surface temperature (Fig. 8e) associated with a cooling rate of -0.53 °C h$^{-1}$, negative $\alpha_{eq}^{closure}$ (-0.69),
low LWP (6.1 g m$^{-2}$) (Fig. 8g), low turbulence (TKE = 0.06 m$^2$ s$^{-2}$ and $\sigma_w^2$ = 0.002 m$^2$ s$^{-2}$), and
negative SHF (-1.7 W m$^{-2}$). As in cases 1 and 2, these atmospheric characteristics allow to maintain
thermally-stable conditions.

Figure 8a indicates that the transition stable/adiabatic fog was observed between 23:00 UTC

and 02:30 UTC (03:30 duration). The vertical profiles of radar reflectivity in Fig. 7a are consistent
with the temperature profiles on the fog stable/adiabatic transition time. The transition time
corresponds with the increase in height and intensity of the radar reflectivity. The fog transition is
observed when on average, the visibility is minimum at Charbonnière (185 m), with low and
negative cooling rate (-0.08 °C h$^{-1}$), low and negative $\alpha_{eq}^{closure}$ (-0.21) which are associated a low
LWP (12.74 g m$^{-2}$) and a RLWP reaching (+10 g m$^{-2}$) (Table 2 and Fig. 8g). These characteristics of
the transition estimated by the fog conceptual model are not consistent with those found by Toledo
et al., 2021, but agree with the vertical profiles of temperature from the MWR (Fig. 8a) and the
increase in turbulence (TKE = 0.1 m$^2$ s$^{-2}$ and $\sigma_w^2$ = 0.008 m$^2$ s$^{-2}$) and SHF (-0.21 W m$^{-2}$) in the fog
layer (Fig. 8f) due to a brisk change in wind direction and speed (Fig. 8d). In summary, the
transition is driven by mechanical turbulence.

The fog adiabatic phase is observed from 02:30 UTC to 03:40 UTC (1 h 10 min duration) at

the supersite under clear sky above the fog top. It is characterized by a decrease of the temperature
inversion top height , of the RLWP (3.45 g m$^{-2}$) and the SHF (-0.49 W m$^{-2}$), and an increase of the
LWP (30.7 g m$^{-2}$), $\alpha_{eq}^{closure}$ (0.54), and cooling rate (0.81 °C h$^{-1}$), while turbulence is kept constant.
The vertical wind shear in the fog top height (Fig. 8d) generates dynamical instability driving the
vertical mixing that reduces the temperature inversion above the fog top (Fig. 8c) which promotes
the vertical development of the fog layer.

A sustainable dissipation is observed at 03:40 UTC. Figure 8d indicates that the dissipation

time is associated with an increase of the wind regime (8 m s$^{-1}$) from the southeast in the entire low-
level atmospheric column attesting the arrival at the supersite of an advected air mass. This front
carried a warm air mass which increased rapidly the near surface temperature (1.34 °C h$^{-1}$) and
allowed a deepening of the fog layer (see Fig. 8c). Advected air mass warms the fog layer causing
the evaporation of the fog water droplets and the lifting of the water vapor by the vertical mixing
driven by turbulence (TKE = 0.42 m$^2$ s$^{-2}$ and $\sigma_w^2$ = 0.07 m$^2$ s$^{-2}$). Thus, the combination between the
decrease in RLWP (2.03 g m$^{-2}$) and SHF (-3.02 W m$^{-2}$), the increase in $\alpha_{eq}^{closure}$ (0.6), surface



temperature (coupling between surface and fog), and turbulence, and a brisk wind allows the mixing
of fog layer with dry air above resulting to the evolution as a stratus . The fog dissipation phase is
thus driven by the advection of warm air at the supersite.

**3.4 Case study 4 (IOP 14) analysis**

As in case study 3 (Fig 7a), the time-cross section of radar reflectivity in Figure 9a indicates
that the water vapor in the lowest atmosphere started to condensate as an ultra-low stratus cloud,
associated with a radiative cooling (IOP 14). Fog formed at the supersite at 21:20 UTC. A stratus
with a base height above the fog top height arrived at around 00:30 UTC corresponding with the fog
vertical extension up to 200 m a.g.l. This cloud is advected from the northwest of the region and is
captured by Meteosat Second Generation (MSG2) (not shown). The first fog dissipation time is
observed at 04:00 UTC. Figure 9b shows that middle-altitude clouds are also observed at the
supersite at around 06:20 UTC. These intermittent clouds contribute to the sustainable dissipation of
the fog at 07:00 UTC by the lifting of its base height. The maximum fog thickness of 300 m is
observed at around 06:00 UTC. In Figure 8c, the time evolution of the visibility at the five sites
shows that the ftime of fog formation shows a shift from west to east, such as in case study 3.
Surface temperatures are contrasted between sites after fog formation and become similar at 04:00
UTC. From midnight to the fog dissipation time, the near surface wind is also the same at all the
sites and blown southerly with intermittent pulses. For the analysis of the processes involved in the
evolution of this case study, we consider its evolution until its first dissipation at 04:00 UTC.
At the supersite, the fog pre-onset phase is characterized by a radiative cooling favoring the
formation of a temperature inversion (Fig. 10c), the occurrence of a westerly wind (Fig. 10d)
transporting mild and wet air from the Atlantic Ocean. The vertical wind shear created by the
increase in wind reduces the intensity of the temperature inversion linked to the radiative cooling (-
0.48 °C h$^{-1}$) (Fig. 10a); negative and low SHF (-1.17 W m$^{-2}$); low turbulence (TKE = 0.06 m$^2$ s$^{-2}$ and
$\sigma_w^2$ = 0.002 m$^2$ s$^{-2}$) and allows the condensation of water vapor in the very low layers driving the
triggering of the ultra-low stratus being the fog. For this episode, the occurrence of middle and high
clouds and the increase in wind at the supersite attests that the fog pre-onset phase is driven by the
advection and radiative cooling as observed in case study 3.
Fog stable phase is observed at the supersite from 21:20 UTC to 23:30 UTC (2 h 10 min
duration) under cloud-free conditions above the fog. It is characterized by a low surface horizontal
visibility (230 m), a negative $\alpha_{eq}^{closure}$ (-0.46), a high cooling rate (-0.88 °C h$^{-1}$), a stable temperature





inversion with 210 m thickness, low LWP (11.34 g m$^{-2}$), negative SHF (-3.26 W m$^{-2}$) and low
turbulence (TKE = 0.09 m$^2$ s$^{-2}$ and $\sigma_w^2$ = 0.012 m$^2$ s$^{-2}$) (see Table 2 and Fig. 10).

The transition between stable and adiabatic fog is observed from 23:30 UTC to 01:00 UTC

(1 h 30 min duration) (see Table 2). As in the previous case studies, this phase is well characterized
by the vertical profiles of temperature and radar reflectivity (Fig. 10a and 10b, respectively) as well
as the rapid increase of $\alpha_{eq}^{closure}$ (from -1.0 to +0.5), a positive RLWP (+11.93 g m$^{-2}$) associated with
increasing LWP (21.19 g m$^{-2}$), moderate turbulence (TKE = 0.19 m$^2$ s$^{-2}$; $\sigma_w^2$ = 0.03 m$^2$ s$^{-2}$), low and
negative SHF (-1.52 W m$^{-2}$) and positive cooling rate (+0.12 °C h$^{-1}$) (Table 2). The fog thickness at
that time is 209 m and the visibility 249 m. Therefore, the transition phase is driven by the
mechanical turbulence produced by the brisk horizontal wind at the supersite (Fig. 10d). The
vertical shear associated with the wind allows a vertical mixing in the fog layer contributing to
reduce the temperature inversion. Note that the brisk wind is associated with the arrival of the
stratus above the fog top height (Fig. 9a and 10b).

At the supersite, fog adiabatic phase is observed from 00:20 UTC to 04:00 UTC (03:40

duration) during this case study. This phase includes a partial dissipation of the fog from 04:00 to
05:30 UTC. The first part of this phase is marked by an increase of the surface horizontal visibility
(372 m), the deepening of the fog layer (CTH = 292 m) and the arrival of an advected stratus cloud.
This period is characterized by episodic brisk winds of southerly flow (Fig. 9d). These episodic
brisk winds are associated with intermittent turbulence (TKE = 0.22 m$^2$ s$^{-2}$ and $\sigma_w^2$ = 0.03 m$^2$ s$^{-2}$),
weak temperature inversion, warming of surface layer (positive cooling rate (+0.47 °C h$^{-1}$)), weak
positive SHF (1.2 W m$^{-2}$), positive RLWP (+8.10 g m$^{-2}$), and high LWP (43.02 g m$^{-2}$). These
characteristics allow the fog to remain optically thick (see Table 2), as in case study 1 and 2.

As in case study 3, the partial nocturnal dissipation of the fog is observed at 04:00 UTC for

this episode. It is characterized by a negative cooling rate (-0.14 °C h$^{-1}$), a slight decrease in LWP
(39.74 g m$^{-2}$) and SHF (0.82 W m$^{-2}$), negative RLWP (-2.32 g m$^{-2}$), moderate turbulence (TKE =
0.27 m$^2$ s$^{-2}$ and $\sigma_w^2$ > 0.04 m$^2$ s$^{-2}$), $\alpha_{eq}^{closure}$ = +0.6, and brisk wind at the supersite (Fig. 10d). This
brisk wind is associated with an increase of the turbulence in the upper levels of the fog layer due to
wind shear. The RLWP indicates that the fog conceptual model estimates fog dissipation time at
04:00 UTC (Fig. 10a) which is consistent with the horizontal visibility (more than 1000 m) and the
maximum value of $\alpha_{eq}^{closure}$. These characteristics indicate that the first fog dissipation processes are
driven by an advection of southern flow at the supersite, as in case study 3.



## 4. Discussion


Figure 11 shows for each fog phase, the mean vertical profiles of air temperature from the

MWR and radar reflectivity from the cloud radar. It highlights the thermal characteristics of fog
phases and differences in atmospheric conditions between fog categories: radiation and radiation-
advection fogs.

For radiation fog case studies (1 and 2), the atmospheric conditions preceding (two hours

before) fog formation are dominated by a strong and thick temperature inversion (more than 14 °C
and 1000 m) which is associated with anticyclonic conditions over Europe favoring easterly wind
and clear sky across the studied area. These atmospheric conditions allow a strong surface radiative
cooling, negative heat fluxes and cooling of near surface air at a rate of -0.9 and -0.7 °C h$^{-1}$ for case
study 1 and 2, respectively. This cooling is associated with low turbulence indicated by low values
of TKE (0.18 m$^2$ s$^{-2}$ in case 1, and 0.06 in case 2) and near surface vertical velocity variance ($\sigma_w^2 <$
0.003 m$^2$ s$^{-2}$) which reinforce the surface thermally stable boundary layer (Fig. 11a and 11b)
favoring the triggering of radiation fog. These results are consistent with the definition of radiation
fog proposed by Price, 2019.

In advection-radiation fog case studies (3 and 4), two hours before fog formation, a westerly

sea breeze is present, transporting mild wet air from the ocean. Surface heat fluxes are negative,
favoring cooling of the near-surface air (-1 °C h$^{-1}$ in case study 3 and -0.5 °C h$^{-1}$ in case study 4) and
turbulent mixing is low (TKE < 0.06 m$^2$ s$^{-2}$). An East-West gradient of formation and dissipation is
observed in line with the westerly synoptic advection of Atlantic inflow. Fog forms earlier in the
West and dissipates later in the East. The combination of advection and radiative cooling favors fog
formation at about 150 m a.g.l as an ultra-low stratus cloud followed by a rapid (less than 30 min)
lowering of the stratus to the surface triggering the onset of the fog in an unstable (case 3) and
neutral (case 4) surface atmospheric boundary layer (Fig. 11c and 11d).

The stable phase is characterized by a stable temperature profile and radar reflectivity which

is maximum near the surface and decreases with height (see Fig. 11). The fog remains shallow (less
than 100 m) with a low LWP ranging less than 12 g m$^{-2}$ proportional to fog depth (Table 2). The
equivalent fog adiabaticity by closure parameter ($\alpha_{eq}^{closure}$) is typically negative during the stable
phase indicating that the fog is not in an adiabatic phase. The near-surface temperature decreases
very moderately (-0.2 °C h$^{-1}$) in cases 1 and 2, while the air keeps cooling at about -1 °C h$^{-1}$ in cases
3 and 4. For the four cases, surface heat fluxes are slightly negative (-3 to 0 W m$^{-2}$) and turbulence
remains low (TKE at about 0.1 m$^2$ s$^{-2}$ and $\sigma_w^2$ at 0.01 m$^2$ s$^{-2}$). This phase is characterized by very low



LWPs (1-2 g m$^{-2}$ for radiation fogs and 6-11 g m$^{-2}$ for advection-radiation fog). For radiation fog
cases, the stable phase lasts around 6 and 4 hours, respectively, while for advection-radiation cases,
it lasts around 2 hours. This is consistent with the strength of the surface inversion of each category
of fog, as shown in Figure 10. These macrophysical characteristics of the fog stable phase are
consistent with those found by Toledo et al., 2021.

The transition from stable to adiabatic phases is a key period in the fog life cycle. This

period is well characterized using the macrophysical parameters of the conceptual model, namely
the equivalent fog adiabaticity by closure ($\alpha_{eq}^{closure}$) parameter of the fog, the fog geometry (CTH)
and fog LWP. During the transition from stable to adiabatic phases, these three parameters increase
significantly (see Table 2). In particular, $\alpha_{eq}^{closure}$ evolves progressively from negative values towards
+0.5 (Toledo et al., 2021). The transition phase lasts from 01:30 to 03:30, however its timing of
occurrence is unpredictable (case 1 at (05:00 - 07:00 UTC), case 2 (00:00 - 02:00 UTC), case 3
(23:00 - 02:30 UTC), and case 4 (23:30 - 01:00 UTC). During this phase, a change is observed in
static stability from stable profiles to neutral and adiabatic profiles (Fig. 11), while the radar
reflectivity profile presents maximum values near the ground that decrease with height (Fig. 11). In
cases 1, 2 and 4, the transition phase is characterized by an increase in turbulence that can explain
the decrease in thermal stability of the fog layer, either shown in the vertical velocity variance ($\sigma_w^2$
>= 0.02 m$^2$ s$^{-2}$) associated with positive surface heat fluxes (cases 1 and 2), or TKE exceeding 0.3
m$^2$ s$^{-2}$. (cases 2 and 4). In all the cases, the fog LWP increases significantly which allows a more
efficient radiative cooling of the fog layer, hence contributing to the destabilization of the fog layer.
In case 3, the transition phase is not marked by a significant increase in turbulence. The transition is
more progressive than in the other case studies (this phase lasts 03:30), the CTH is only 25 m
deeper on average than during the stable phase, the $\alpha_{eq}^{closure}$ remains low during that phase, and
reaches 0.5 at the end of the transition phase.

According to temperature vertical profiles from the MWR, at the end of the transition time

from stable to adiabatic fog, the temperature profile becomes neutral or slightly unstable. This is
consistent with the definition of the transition given by Price et al., 2011. We also find that it is
during this period that the fog reaches its maximum value of RLWP, showing that the LWP
increases beyond the critical liquid water path value, which gives information on the persistence of
fog.

For radiation fog case studies, the adiabatic phase lasts 04:00 and 06:40 for case 1 and 2

respectively, maintaining the fog life cycle during the night until after sunrise. In cases 3 and 4, the



adiabatic phase is shorter and lasts 01:00 and 03:40, respectively, with a night-time dissipation at 03:40 and 04:00 UTC, respectively. In this fog phase, for radiation fog, the LWP ranges from 22-26 g m$^{-2}$ with CTH near 190 m a.g.l. The fog is deeper for advection-radiation fog cases with LWP / CTH at 30 g m$^{-2}$ / 200 m a.g.l and 43 g m$^{-2}$ / 290 m a.g.l, respectively (Table 2). The adiabatic phase is characterized by an equivalent fog adiabatic by closure parameter near or above 0.5, and a positive but low RLWP. For all the cases except case 3, the adiabatic phase is associated with moderate turbulence in the fog layer ($0.2 < TKE < 0.4$ m$^2$ s$^{-2}$ and $0.03 < \sigma_w^2 < 0.04$ m$^2$ s$^{-2}$) which indicates significant vertical mixing generating an unstable surface atmospheric boundary layer (Fig. 11). This finding is consistent with the result of Ju et al., 2020 who based their analysis on one case study and Ghude et al., 2023, Dhangar et al., 2021 and Zhou and Ferrier, 2008 for more case studies analysis. In addition, this phase can also be driven by horizontal advection (mesoscale and synoptic systems) as in the case study 3.

This study shows two fog dissipation periods, at night and after sunrise. Daytime dissipation is observed for radiative fog cases and night-time dissipation for advection-radiation ones. All of them are observed when $\alpha_{eq}^{closure} > 0.5$, $TKE > 0.3$ m$^2$ s$^{-2}$, $\sigma_w^2 > 0.04$ m$^2$ s$^{-2}$, and the LWP $> 40$ g m$^{-2}$ (except case study 2). For cases 1 and 2, turbulence is thermally driven by positive SHF, while for cases 3 and 4, the night-time turbulence increase is mechanically driven by increased wind speed. For all cases, the RLWP decreases significantly from the stable phase to the dissipation phase, confirming that dissipation through fog-base lifting is linked to insufficient liquid water content in the fog layer, as suggested by the conceptual model. For case 3, the RLWP becomes negative 20 min after dissipation. This delay is likely due to very rapid changes in LWP and CTH at the time of dissipation.

## 5. Summary and Conclusions

The SOFOG3D field campaign provided a unique dataset documenting thermodynamic and dynamical atmospheric circulations to further understand the processes driving fog formation and dissipation over Southeastern France. Based on an innovative instrumental synergy combining in-situ and remote sensing measurements gathered in an adiabatic fog conceptual model, this study has documented the processes favoring fog evolution. The analysis has focused on four fog case studies: two radiative and two advective-radiative fogs. For each case study, we have defined the different phases characterizing the fog life cycle, namely (i) its formation, (ii) an initial phase where the fog develops under thermally stable conditions, (iii) a transition phase towards an adiabatic fog,



(iv) an adiabatic phase during which the fog vertical profile is adiabatic, and (v) a dissipation phase
where the fog base lifts.
The results showed that for both radiation fog cases, the conditions are marked by very cold
atmospheric conditions associated with a continental easterly nocturnal low-level jet. For these
cases, the stable fog phase develops under weak turbulence and strong surface radiative cooling and
strong and deep surface temperature inversion layer. The transition phase is driven by an increase in
turbulence in the fog layer. This turbulence is associated with a change in the air mass
thermodynamical characteristics by advection. The adiabatic phase is observed when the turbulence
$(0.2 < \text{TKE} < 0.4\ \text{m}^2\ \text{s}^{-2})$ is sufficient to ensure vertical mixing in the fog layer. For these fog events,
dissipation time is observed when the thermal and dynamic production of the turbulence are high
$(\text{TKE} > 0.4\ \text{m}^2\ \text{s}^{-1}\ \text{and}\ \sigma_w^2 > 0.04\ \text{m}^2\ \text{s}^{-2})$. For this category of fog, the adiabatic fog conceptual model
estimates the dissipation time one hour before its observation.
The analysis on the advection-radiation case studies shows that they have the shortest life
cycle linked to the low surface boundary layer stability due to the vertical mixing generated by the
westerly strong wind. In this category of fog, the processes driving the stable, stable/adiabatic
transition and adiabatic phases are similar to those of the radiation fog category. However, the
dissipation phase is driven by night-time horizontal advection at the supersite.
In summary, LWP and RLWP measured during SOFOG3D present lower values than at the
SIRTA site, close to the uncertainty of the measurement. The conceptual model has therefore
difficulties in integrating the mixing phases in the fog layer. Further development of the model is
needed to adapt it to other regions before it can be used for nowcasting prediction. Fog formation,
evolution and dissipation across southern France require an analysis of the synoptic atmospheric
circulation in terms of wind, cloud cover, and thermodynamical processes. Indeed, this paper
highlights that fog nowcasting in this region needs in addition to the numerical weather prediction
models, a cloud radar, a microwave radiometer, a wind lidar, a surface energy balance, and
meteorological stations. Operationalizing these instruments would allow to improve fog
nowcasting, which will reduce its socioeconomic impacts in this region.
**Appendix A: Fog conceptual model parametrization**
**A.1 Liquid water content**
The conceptual model for adiabatic fog has been developed at SIRTA by Toledo et al.,
2021. This model is a unidimensional model inspired by previous numerical models for stratus





clouds (Betts, 1982, Albrecht et al., 1990; and Cermak and Bendix, 2011) (see equation 1). The
basic hypothesis is to consider a well-mixed fog layer and express the increase with height of the
fog liquid water content as a function of the local adiabaticity ($\alpha(z)$) and the negative of the change
in the saturation mixing ratio with height ($\Gamma_{ad}(T,P)$), given in equation A1.
$$\frac{dLWC(z)}{dz} = \alpha(z)\Gamma_{ad}(T,P) \qquad\qquad\qquad (A1)$$

Where T and P are air temperature and pressure, respectively. z is the height above the

surface and varies between 0 and the cloud top height (CTH). By integrating equation 1, it is
important to take into account fog geometry which is different from that of the stratus cloud. For a
fog, the LWC at the base is non-zero due to the presence of liquid droplets down to the ground
level. This presence of droplets drives surface visibility reduction and water deposition on the soil.
Thus, as indicated in equation A2, the vertical integral of the LWC(z) is a function of the variation
with height of the adiabaticity, $\Gamma_{ad}(T,P)$ and the measurement of the LWC at surface ($LWC_0$). This
equation shows that the LWC increases with the thickness of the fog up to the height where upward
motions of moisture from the surface are constrained by downward motions of dry air from the fog
top height (Walker, 2003; Cermak and Bendix, 2011). From this interface level, the LWC decreases
with height and becomes zero at the fog top height (Brown and Roach, 1976; Cermak and Bendix,

2011).

$$LWC(z) = \int_{z'=0}^{z'=z} \alpha(z')\Gamma_{ad}(T,P)\,dz' + LWC_0 \qquad\qquad (A2)$$

## A.2 Liquid water path

The fog liquid water path (LWP) represents the total amount of liquid water present in the

fog layer. It can be estimated by integrating equation A2 in height considering that the fog thickness
is equivalent to the CTH (equation A3). An approximation assuming a constant adiabaticity is
introduced by using the equivalent fog adiabaticity term $\alpha_{eq}$. This simplifies the calculation, since a
complete computation would require a knowledge of the vertical profile of adiabaticity which
depends on the thermodynamic properties of the fog layer. In this conceptual model, the LWC is
treated as if it increased linearly with height from the surface to the CTH. At the surface level the
LWC from the model and fog are the same, connecting a given LWP with surface LWC. This
quantity is converted to visibility values using Gultepe et al., 2006 parametrization. Hence, the



conceptual model connects fog LWP with its CTH and surface visibility values, it provides an

estimation of the equivalent fog adiabaticity.

$$LWP = \frac{1}{2} \alpha_{eq} \Gamma_{ad}(T, P) CTH^2 + LWC_0 CTH \qquad (A3)$$

**A.3 Critical liquid water path**

Considering that the fog dissipates when its liquid water path is below a certain threshold

depending on the local thermodynamic atmospheric conditions. In case of dissipation by lifting the

base height of the fog, Wærsted, 2018 found a deficit in LWP in the fog layer. This assertion allows

defining a minimum amount of LWP necessary to maintain the horizontal visibility at surface lower

or equal to 1000 m, defined as the critical liquid water path (CLWP). Thus, based on equation A3,

the CLWP can be expressed in equation A4 considering a critical liquid water content at surface

(LWCc). Theoretically, the LWCc is the LWC that would cause a 1000 m visibility. It is estimated

from the parameterization of Gultepe et al., 2006 based on the horizontal visibility at surface.

$$CLWP = \frac{1}{2} \alpha_{eq} \Gamma_{ad}(T, P) CTH^2 + LWC_c CTH \qquad (A4)$$

*Data availability.* All the data used in this study are hosted by the the French national center for

Atmospheric data and services AERIS in the link https://sofog3d.aeris-data.fr/catalogue/#masthead.

Data access can be free following the conditions fixed by the SOFOG3D project.

*Competing interests.* The authors claim no conflict of interest for this study.

*Author contributions.* **Cheikh DIONE:** Conceptualization, Methodology, Investigation,
Validation, Formal-analysis, Writing – original draft, Writing – review & editing, Visualization.
**Martial HAEFFELIN:** Supervision, Methodology, Investigation, Formal-analysis, Writing-
original draft, Writing-review & Editing, , Funding acquisition. **Jean-Charles DUPONT:**
Supervision, Investigation, Editing. **Felipe TOLEDO:** Methodology, Investigation, Editing.
**Frederic BURNET:** Project administration, Resources, Investigation, Editing, Funding acquisition.
**Christine LAC:** Supervision, resources, Investigation, editing, Funding acquisition. **Jean-Francois
RIBAUD:** Visualization, Investigation, Editing. **Pauline MARTINET:** Editing, Investigation,
Resources, Data curation. **Guylaine CANUT:** Investigation, Editing, Data curation. **Susana
JORQUERA:** Data curation. **Julien DELANOË:** Data curation.



*Acknowledgments.* The SOFOG3D field campaign was supported by METEO-FRANCE and ANR through grant AAPG 2018-CE01-0004. Data are managed by the French national center for Atmospheric data and services AERIS. The CNRM/GMEI/LISA team supported the deployment, monitoring and data processing and supplying of Wind lidar and microwave radiometer.

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





**List of tables**

**Table 1 :** Case study number, fog onsets, type of fog formation, fog dissipation times, fog duration and type of fog dissipation for the four documented case studies. Time is in UTC. Dates are in the format "dd/mm/yyyy". "dd" indicates the day, "mm" the month, and "yyyy" the year.

**Table 2 :** Summary of fog features at the supersite during the five defined phases during its evolution for each case study. The formation, dissipation times are estimated using the visibility (m) from the Scatterometer. The transition from stable to adiabatic fog is defined using temperature from the microwave radiometer. The cooling rate (dT/dt), wind speed (WS), and wind direction (WD) are derived from the meteorological station. Sensible heat flux (SHF), turbulent kinetic energy (TKE) and the vertical velocity variance ($\sigma_w^2$) at 3 m a.g.l are derived from the flux station. The liquid water path (LWP) is estimated from the MWR. The fog reservoir of liquid weather path (RLWP) and the equivalent adiabaticity of closure $\alpha_{eq}^{closure}$ parameter are computed by the conceptual model. Fog top height (FTH) and middle and high cloud base and top heights are derived from the radar reflectivity from Basta cloud radar."-" indicates that the variables are not measurable or calculable.



**List of figures**

**Figure 1:** In a), the geographical map of the study area of the SOFOG3D field campaign including the five instrumented sites (Agen, Bergerac, Biscarrosse, Mont-de-Marsan, and Saint-Symphorien) where a microwave radiometer was installed. Blue lines indicate the rivers. The cities are indicated in black dots. The most instrumented domain around the supersite is indicated in a) by the red rectangle. In b), the orography of a 100 x 100 km² domain centered on Charbonnière which includes locations of four of the meteorological stations installed around the supersite used in this study. Orography data are from the National Aeronautics and Spatial Administration (NASA) shuttle radar topography mission (SRTM) (90 m of resolution).

**Figure 2: Figure 2:** (a) Scatter plot of the equivalent adiabaticity by closure versus the CTH and LWP at the supersite. b) Boxplot of the equivalent adiabaticity by closure versus the different LWP ranges from the MWR. In b), numbers at the figure top indicate total values included in each boxplot and computed between 2 hours before and after the fog. Horizontal dashed line indicates the threshold of the equivalent adiabaticity from closure defining the transition from stable to adiabatic fog.

**Figure 3:** In (a-b) time-height cross-section from surface up to 600 and 12000 m, respectively of radar reflectivity from Basta (shaded) radar, time evolution of the cloud top height from Basta (red line), and the cloud base height from the Celiometer (CL51) (green line). Time evolution of (c) surface visibility, (d) 10 m wind speed, (e) 2 m air temperature, and (f) 10 m wind direction observed on the 28-29 December 2019 (case study 1, IOP 5) at the five meteorological stations (in red, black, blue, green, and pink lines for Moustey (1 m a.g.l), Charbonnière (3 m a.g.l), Cape Sud (3 m a.g.l), Tuzan (3 m a.g.l), and Noaillan (1 m a.g.l), respectively) deployed around the supersite. Note that wind was not collected at Tuzan. In (c), the visibility measured at Moustey was interrupted by technical issues. Vertical black dashed lines indicate fog formation (left) and dissipation (right) times. Green dashed lines show the transition time from stable fog to adiabatic fog (fog mature phase). Red dashed line indicates the sunrise.

**Figure 4:** Evolution of fog macrophysical characteristics observed on the 28-29 December 2019 (case study 1, IOP 5) at Charbonnière. In (a-b) vertical profiles of air temperature from the Hatpro microwave radiometer (MWR) and radar reflectivity from Basta radar, respectively. In (c) time-



height cross-section of air temperature from the MWR (shaded), time evolution of inversion top height (ITH) (open gray circles), inversion base height (IBH) (open gray squares), cloud top height (CTH) from the cloud radar (open black squares), and the cloud base height (CBH) from the Celiometer (open black circles). In (d) wind speed (shaded) and direction (arrows) from the WindCube. Arrows in (d) indicate only the direction of the horizontal flow. Time evolution of (e) air temperature at 3 m a.g.l from the meteorological station (red line) and equivalent adiabaticity of closure from the fog conceptual model (blue line), (f) the mean of the turbulent kinetic energy (TKE) in the layer 40 – 220 m for the WindCube (black line) and the TKE (blue line) and vertical velocity variance (red line) at 3 m a.g.l from the flux station at Charbonnière, (g) the LWP estimate from the MWR (blue line), the RLWP from the fog conceptual model (red line), and (h) sensible heat fluxes (SHF) (red and blue lines, respectively) from the flux station. Vertical black dashed lines indicate fog formation and dissipation times. Green dashed lines indicate the transition period (fog mature phase) from stable to adiabatic fog. The red dashed line indicates sunrise.

**Figure 5:** As in Figure 3 but for the 5-6 January 2020 (case study 2, IOP 6). In (c), only Charbonnière and Noaillan have valid data. In (c), the visibility measured at Moustey, Tuzan and Cape Sud were interrupted by technical issues.

**Figure 6:** As in Figure 4 but for the 5-6 January 2020 (case study 2, IOP6).

**Figure 7:** As in Figure 3 but for the 8-9 February 2020 (case study 3, IOP 11).

**Figure 8:** As in Figure 4 but for the 8-9 February 2020 (case study 3, IOP 11).

**Figure 9:** As in Figure 3 but for the 7-8 March 2020 (case study 4, IOP 14).

**Figure 10:** As in Figure 4 but for the 7-8 March 2020 (case study 4, IOP 14). The LWP, RLWP, and $\alpha_{eq}^{closure}$ are disrupted between 00:30 and 02:30 UTC because the LWP estimated by the MWR take into account the liquid water in the advected stratus.

**Figure 11:** Vertical profiles of air temperature and radar reflectivity put together for each fog case study: (a) for case study 1, (b) case study 2, (c) case study 3 and (d) case study 4. Line and shaded



area indicate the mean and standard deviation of air temperature and radar reflectivity during each
fog phase.





***Table 2 :*** *Summary of fog features at the supersite, during the five defined phases during it evolution for each case*
*study. The formation, dissipation times are estimated using the visibility (m) from Scatterometer. The cooling rate*
*(dT/dt), wind speed (WS) and wind direction (WD) are derived from the meteorological station. Sensible heat flux*
*(SHF), turbulent kinetic energy (TKE), and the vertical velocity variance ($\sigma_w^2$) at 3 m a.g.l are derived from the flux*
*station. The liquid water path (LWP) is estimated from the MWR. The fog reservoir of liquid weather path (RLWP) and*
*the equivalent adiabaticity of closure $\alpha_{eq}^{closure}$ parameter are computed by the conceptual model. Fog top height (FTH)*
*and middle and high cloud base and top heights are derived from the radar reflectivity from Basta cloud radar."-"*
*indicates that the variables are not measurable or calculable.*

| Case study number | Phase names | Time range | Duration (h:min) | Visibility (m) | dT/dt (°C h⁻¹) | $\alpha_{eq}^{closure}$ g m⁻³ | LWP (g m⁻²) | RLWP max (g m⁻²) | FTH (m a.g.l) | WS (m s⁻¹) | WD (°) | TKE (m² s⁻²) | $\sigma_w^2$ (m² s⁻²) | SHF (W m⁻²) | Cloud above fog (m a.g.l) |
|---|---|---|---|---|---|---|---|---|---|---|---|---|---|---|---|
| 1 (IOP5) | Pre-fog phase | [20:50 - 22:50] | 2:00 | 9962 | -0.9 | - | 0 | - | - | 0.61 | 61 | 0.18 | 0.002 | -0.23 | clear |
| | Stable | [22:50 - 05:00] | 6:10 | 736 | -0.18 | -1.3 | 2.18 | - | 51 | 0.7 | 84 | 0.12 | 0.01 | -1.16 | clear |
| | Transition stable/adiabatic | [05:00 - 07:00] | 2:00 | 173 – 262 | 0.08 | -0.8 – 0.4 | 7 – 28 | 8 – 15 | 68 – 181 | 0.5 – 2.1 | 68 – 112 | 0.07 – 0.17 | 0.02 – 0.03 | 2.3 – 8.8 | clear |
| | Adiabatic | [07:00 - 11:00] | 4:00 | 370 | 0.77 | 0.5 | 26.16 | 6.38 | 185 | 2.4 | 116 | 0.28 | 0.04 | 12.9 | [8000 - 10000] |
| | Dissipation | [10:30 - 11:30] | 1:00 | 1549 | 1.1 | 0.63 | 43.34 | -11.39 | 288 | 2.6 | 94 | 0.46 | 0.06 | 22.02 | clear |
| 2 (IOP6) | Pre-fog | [18:40 - 20:40] | 2:00 | 15566 | -0.7 | - | 0 | - | - | 0.2 | 195 | 0.06 | 0.003 | -0.17 | clear |
| | Stable | [20:40 - 00:00] | 3:20 | 242 | -0.13 | -0.69 | 1.66 | - | 71 | 1 | 183 | 0.09 | 0.009 | 0.28 | clear |
| | Transition stable/adiabatic | [00:00 - 02:00] | 2:00 | 219 – 291 | -0.007 | -0.2 – 0.45 | 0.3 – 17 | -0.23 – 3.8 | 81 – 168 | 1.6 – 2.6 | 149 – 147 | 0.35 – 0.25 | 0.02 – 0.04 | 3.7 – 11 | clear |
| | Adiabatic | [02:00 - 08:40] | 6:40 | 450 | 0.17 | 0.51 | 22.14 | 1.51 | 191 | 2.2 | 110 | 0.27 | 0.04 | 6.62 | clear |
| | Dissipation | [08:10 - 09:10] | 1:00 | 944 | 0.43 | 0.53 | 11.62 | -7.63 | 187 | 2.5 | 136 | 0.33 | 0.048 | 14.02 | [250 - 1000 ] |
| 3 (IOP11) | Pre-fog | [18:40 - 20:40] | 2:00 | 13239 | -1.03 | - | 0 | - | - | 1.3 | 242 | 0.03 | 0.011 | -5.5 | rain |
| | Stable | [20:40 - 23:00] | 2:20 | 243 | -1.2 | -0.69 | 6.10 | - | 77 | 1 | 220 | 0.06 | 0.002 | -1.7 | clear |
| | Transition stable/adiabatic | [23:00 - 02:30] | 3:30 | 134 – 260 | -0.08 | -1.35 – 0.4 | 5 – 19.8 | 7.7 – 6 | 50 – 156 | 1.8 – 0.4 | 144 – 78 | 0.07 – 0.04 | 0.006 – 0.004 | -1.9 – -0.2 | clear |
| | Adiabatic | [02:30 - 03:40] | 1:10 | 271 | 0.81 | 0.54 | 30.70 | 3.45 | 204 | 1 | 120 | 0.08 | 0.008 | -0.49 | clear |
| | Dissipation | [03:10 - 04:10] | 1:00 | 1445 | 1.34 | 0.6 | 41.90 | 2.03 | 235 | 3.6 | 143 | 0.42 | 0.07 | -3.02 | clear |
| 4 (IOP14) | Pre-fog | [19:20 - 21:20] | 2:00 | 14088 | -0.47 | - | 0 | - | - | 1.1 | 233 | 0.06 | 0.002 | -1.17 | [5000 - 6000] [8000 - 10000] |
| | Stable | [21:20 - 23:30] | 2:10 | 230 | -0.88 | -0.46 | 11.34 | - | 81 | 1.2 | 177 | 0.09 | 0.012 | -3.26 | clear |
| | Transition stable/adiabatic | [23:30 - 01:00] | 1:30 | 240 – 253 | 0.12 | -0.17 – 0.64 | 10.9 – 59.2 | 10 – - | 106 – 209 | 1.6 – 2.7 | 141 – 184 | 0.08 – 0.32 | 0.01 – 0.05 | -1.6 – 2.7 | clear |
| | Adiabatic | [00:20 - 04:00] | 3:40 | 372 | 0.47 | 0.59 | 43.02 | 8.10 | 292 | 2 | 179 | 0.22 | 0.03 | 1.2 | [250 - 500] |
| | Dissipation | [03:30 - 04:30] | 1:00 | 1160 | -0.14 | 0.60 | 39.74 | -2.32 | 240 | 2.7 | 174 | 0.27 | 0.04 | 0.82 | clear |



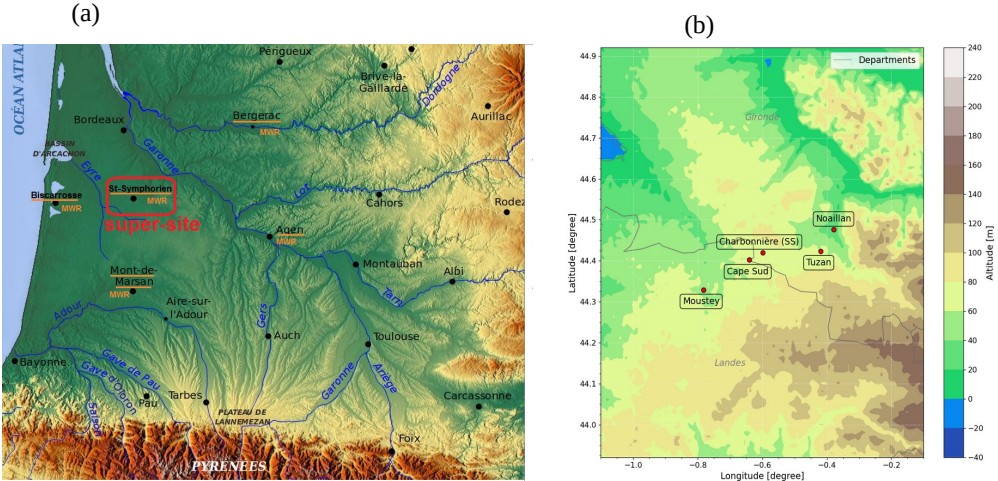

**Figure 1 :** In a), the orography of the study area of the SOFOG3D field campaign including the five instrumented sites (Agen, Bergerac, Biscarrosse, Mont-de-Marsan, and Saint-Symphorien) where a microwave radiometer was installed. Blue lines indicate the rivers. The cities are indicated in black dots. The most instrumented domain around the supersite is indicated in a) by the red rectangle. In b), the orography of a 100 x 100 km$^2$ domain centered on Charbonnière which includes locations of four of the meteorological stations installed around the supersite and used in this study. Orography data are from the National Aeronautics and Spatial Administration (NASA) shuttle radar topography mission (SRTM) (90 m of resolution).



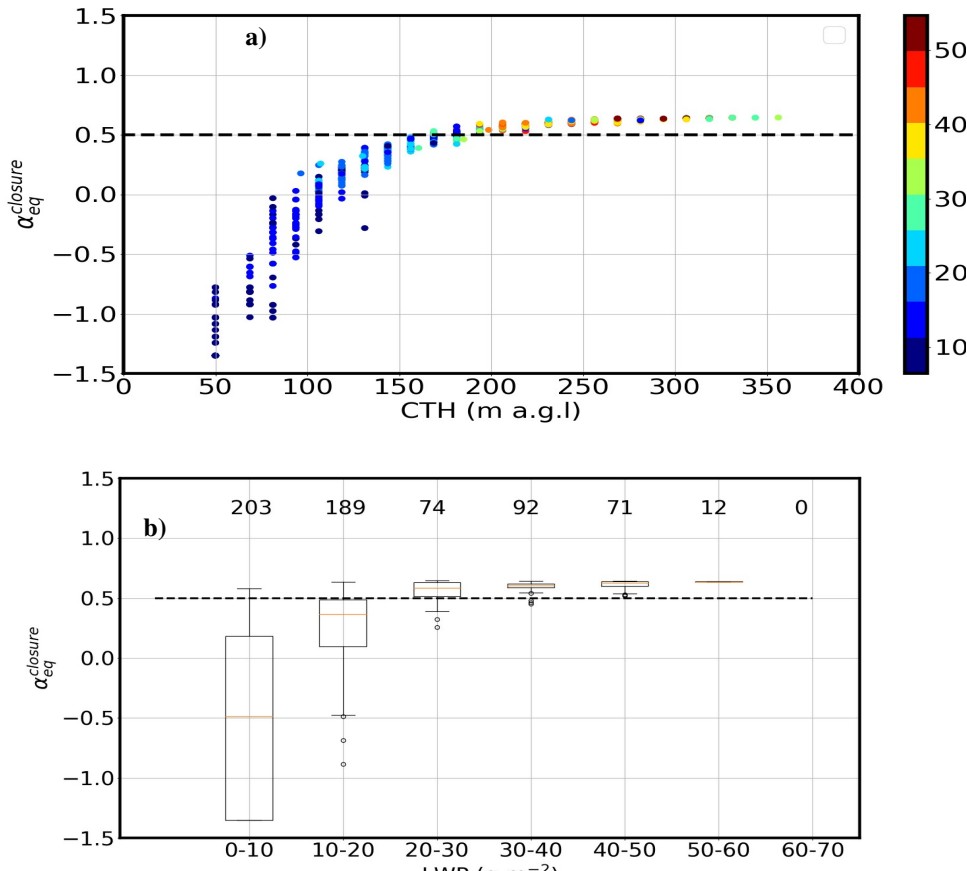

**Figure 2:** In (a) Scatter plot of the equivalent adiabaticity by closure versus the CTH and LWP (colored circles) at the supersite. b) Boxplot of the equivalent adiabaticity by closure versus the different LWP ranges from the MWR. In b), numbers at the figure top indicate total values included in each boxplot and computed between 2 hours before and after the fog. Horizontal dashed line indicates the threshold of the equivalent adiabaticity from closure defining the transition from stable to adiabatic fog.



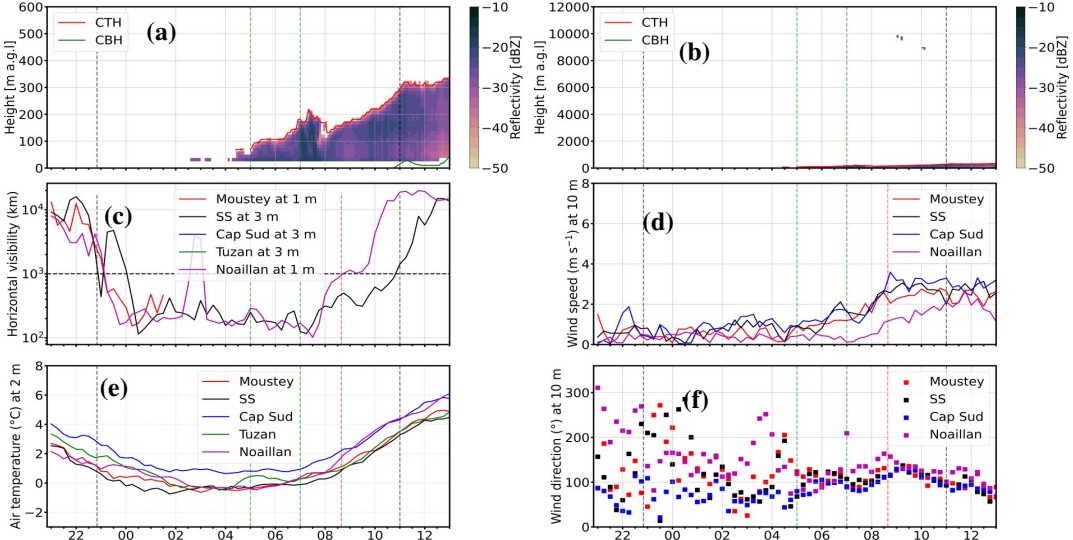

**Figure 3:** In (a-b) time-height cross-section from surface up to 600 and 12000 m, respectively of radar reflectivity from Basta (shaded) radar, time evolution of the cloud top height from Basta (red line), and the cloud base height from the Celiometer (CL51) (green line). Time evolution of (c) surface visibility, (d) 10 m wind speed, (e) 2 m air temperature, and (f) 10 m wind direction observed on the 28-29 December 2019 (case study 1, IOP 5) at the five meteorological stations (in red, black, blue, green, and pink lines for Moustey (1 m a.g.l), Charbonnière (3 m a.g.l), Cape Sud (3 m a.g.l), Tuzan (3 m a.g.l), and Noaillan (1 m a.g.l), respectively) deployed around the supersite. Note that wind was not collected at Tuzan. In (c), the visibility measured at Moustey was interrupted by technical issues. Vertical black dashed lines indicate fog formation (left) and dissipation (right) times. Green dashed lines show the transition time from stable fog to adiabatic fog (fog mature phase). Red dashed line indicates the sunrise.



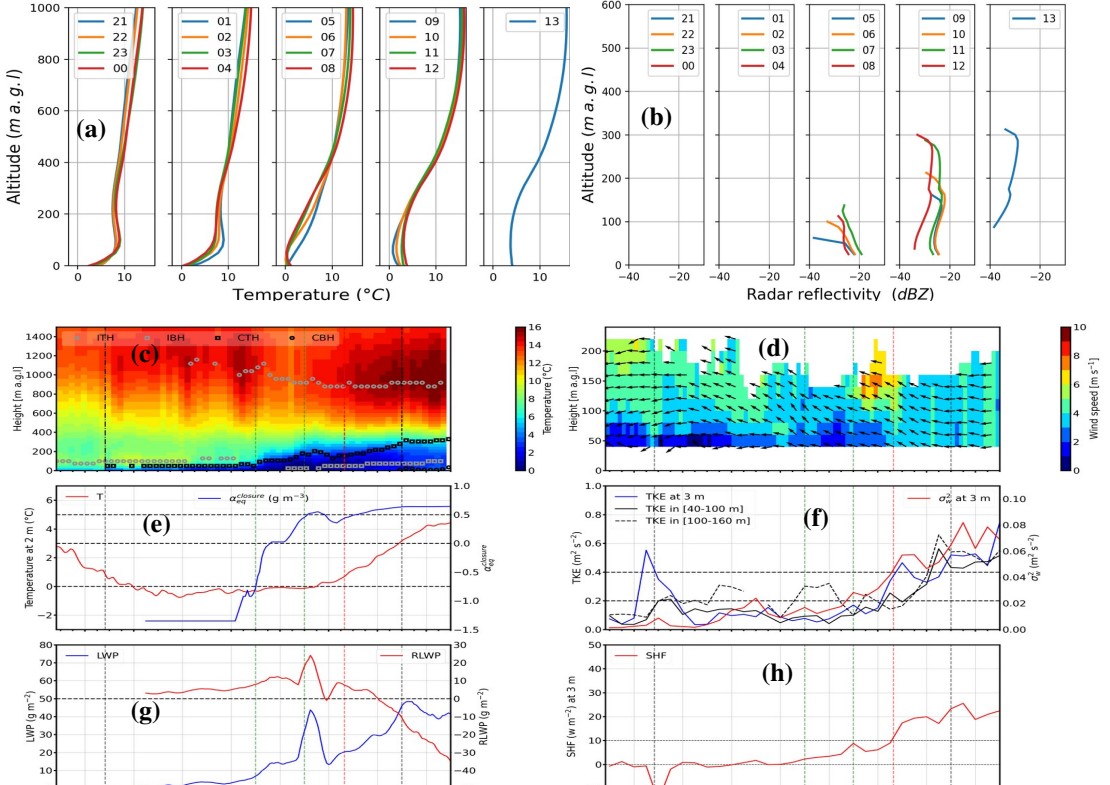

**Figure 4:** Evolution of fog macrophysical characteristics observed on the 28-29 December 2019 (case study 1, IOP 5) at Charbonnière. In (a-b) vertical profiles of air temperature from the Hatpro microwave radiometer (MWR) and radar reflectivity from Basta radar, respectively. In (c) time-height cross-section of air temperature from the MWR (shaded), time evolution of inversion top height (ITH) (open gray squares), inversion base height (IBH) (open gray squares), cloud top height (CTH) from the cloud radar (open black squares), and the cloud base height (CBH) from the Celiometer (open black circles). In (d) wind speed (shaded) and direction (arrows) from the WindCube. Arrows in (d) indicate only the direction of the horizontal flow. Time evolution of (e) air temperature at 3 m a.g.l from the meteorological station (red line) and equivalent adiabaticity of closure from the fog conceptual model (blue line), (f) the mean of the turbulent kinetic energy (TKE) in the layer 40 – 220 m for the WindCube (black line) and the TKE (blue line) and vertical velocity variance (red line) at 3 m a.g.l from the flux station at Charbonnière, (g) the LWP estimate from the MWR (blue line), the RLWP from the fog conceptual model (red line), and (h) sensible heat fluxes (SHF) (red and blue lines, respectively) from the flux station. Vertical black dashed lines indicate fog formation and dissipation times. Green dashed lines indicate the transition period (fog mature phase) from stable to adiabatic fog. The red dashed line indicates sunrise.



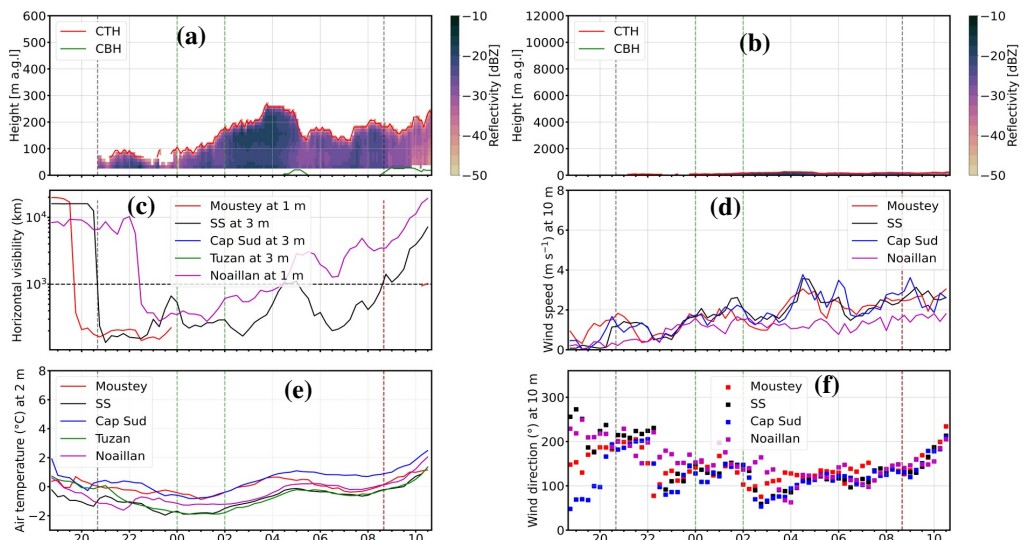

**Figure 5:** As in Figure 3 but for the 5-6 January 2020 (case study 2, IOP 6). In (c), only Charbonnière and Noaillan have valid data. In (c), the visibility measured at Moustey, Tuzan and Cape Sud were interrupted by technical issues.



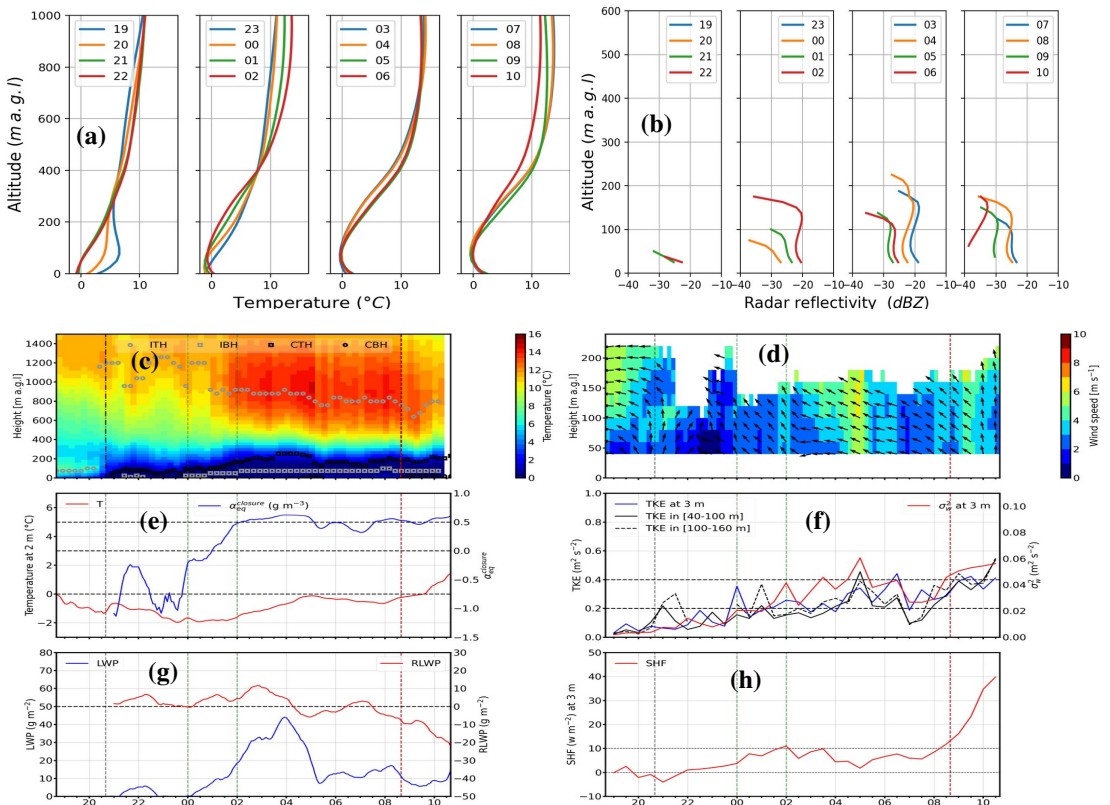

**Figure 6:** As in Figure 4 but for the 5-6 January 2020 (case study 2, IOP6). The red vertical dashed line indicates the sunrise.



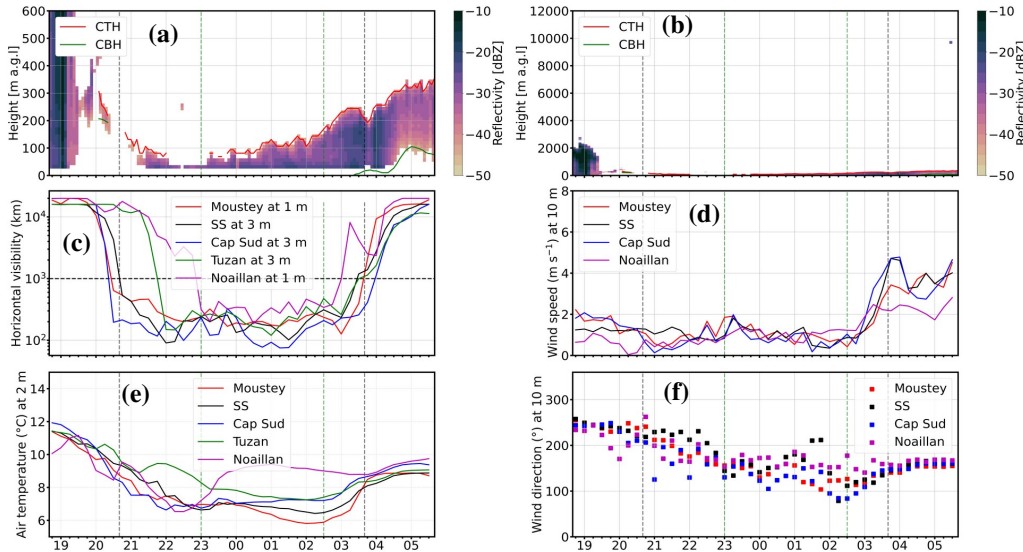

**Figure 7 :** As in Figure 3 but for the 8-9 February 2020 (case study 3, IOP 11)



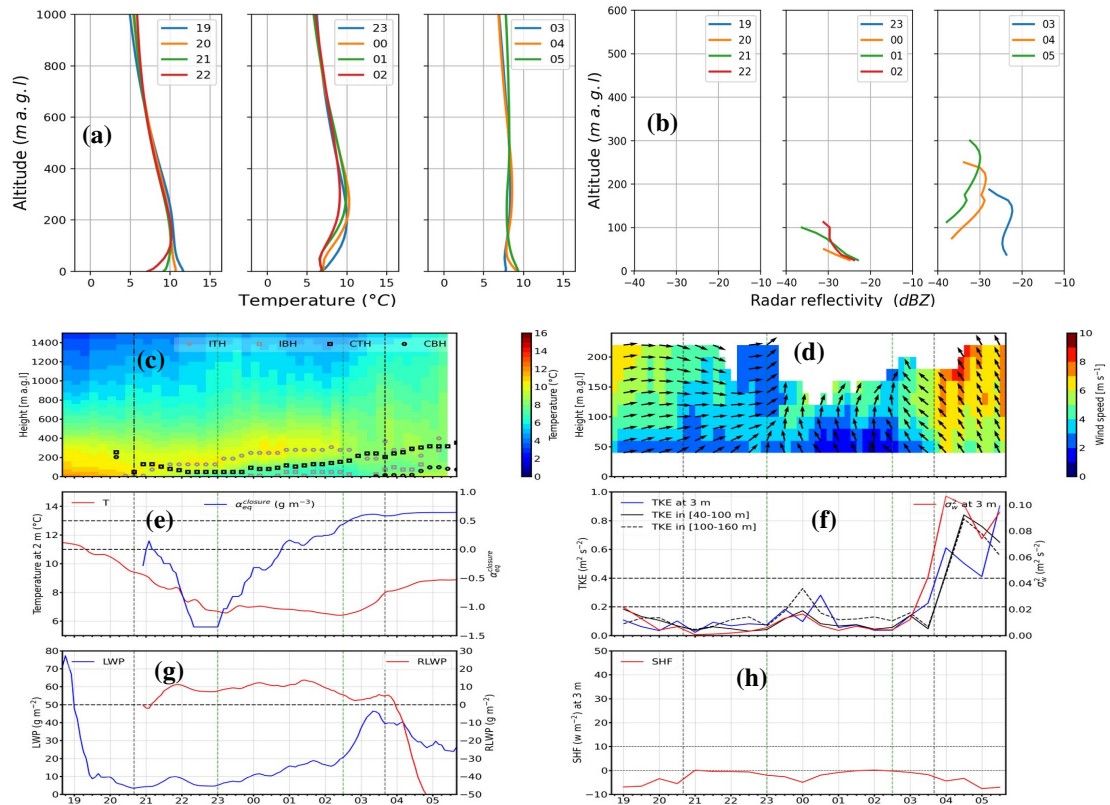

**Figure 8:** As in Figure 4 but for the 8-9 February 2020 (case study 3, IOP 11).





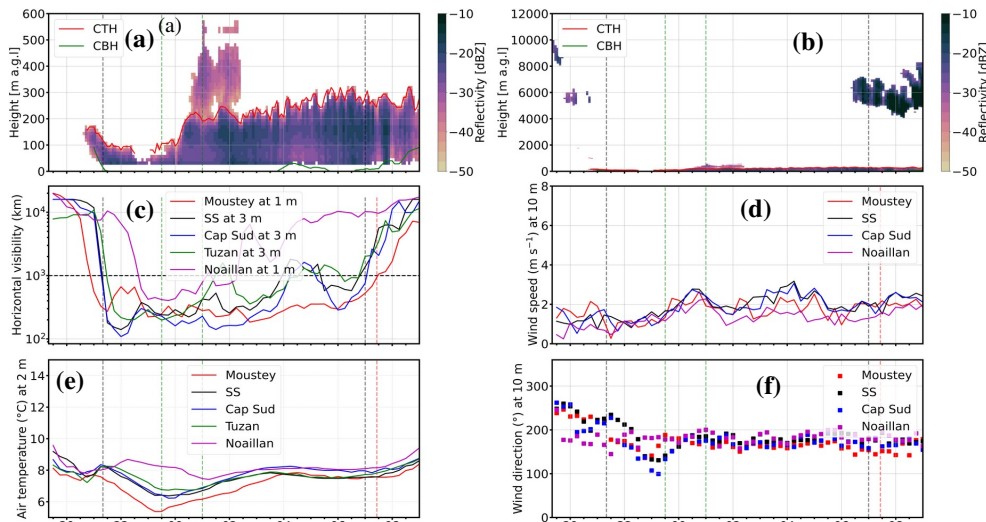

**Figure 9 :** As in Figure 3 but for the 7-8 March 2020 (case study 4, IOP 14).



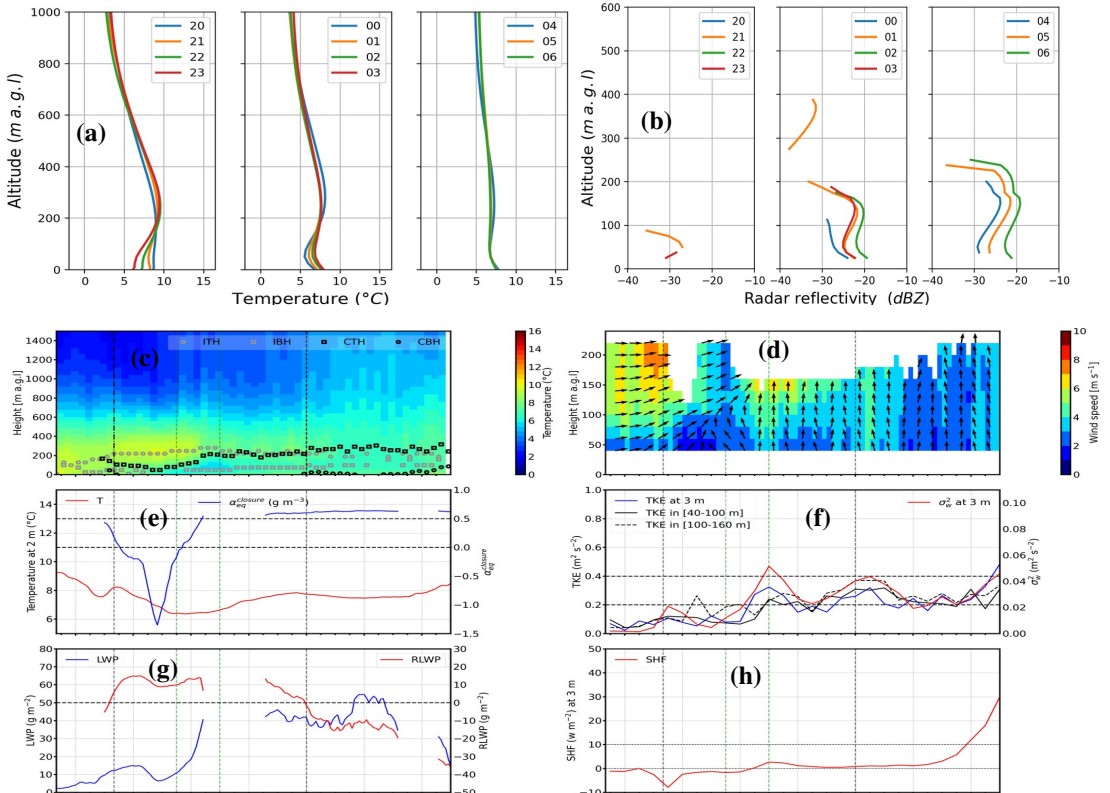

**Figure 10 :** As in Figure 4 but for the 7-8 March 2020 (case study 4, IOP 14). The LWP, RLWP, and $\alpha_{eq}^{closure}$ are disrupted between 00:30 and 02:30 UTC because the LWP estimated by the MWR take into account the liquid water in the advected stratus.





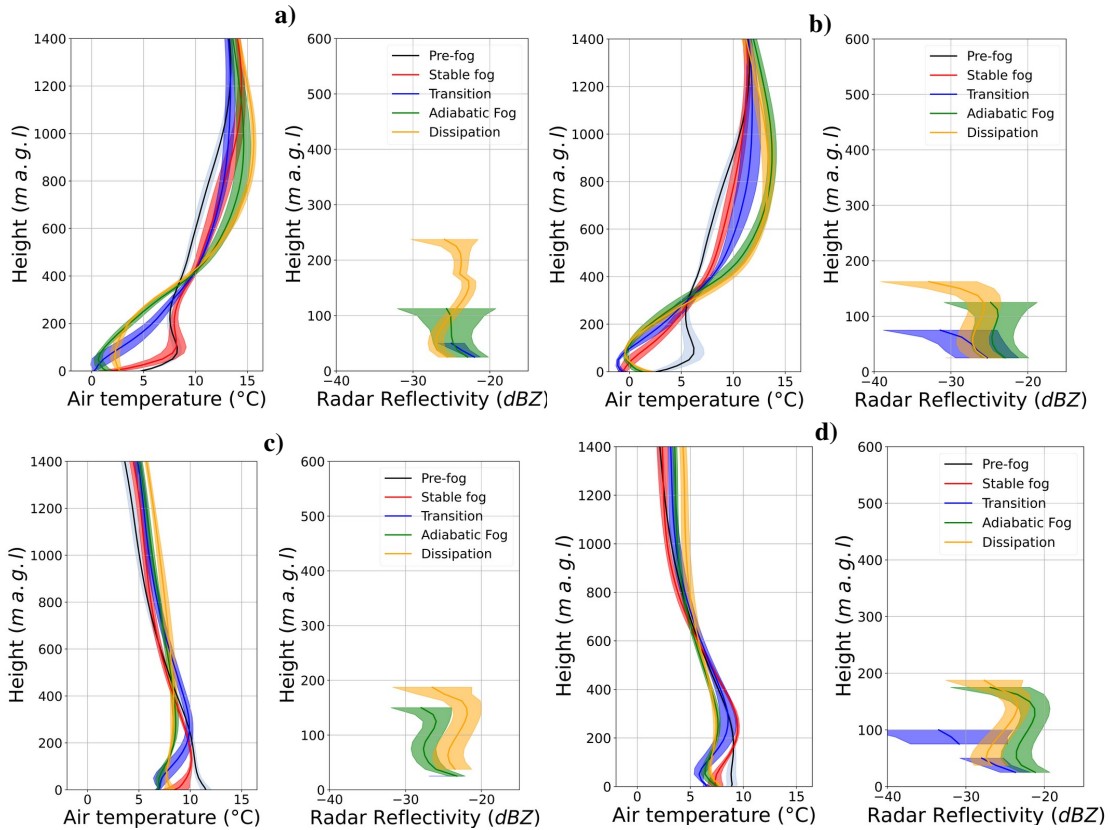

**Figure 11:** Vertical profiles of air temperature and radar reflectivity put together for each fog case study (a) for case 1, (b) case 2, (c) case 3 and (d) case 4. Line and shaded area indicate the mean and standard deviation of air temperature and radar reflectivity during each fog phase.