# Peer review of "SOFOF3D experiment"

_EGUsphere, 2023_

## Referee Comment (RC2)

**Review of egusphere-2023-1224 : Role of thermodynamic and turbulence processes on the fog life cycle during SOFOF3D**

This manuscript presents case studies of four of the heaviest fog events from the SOFOG3D field experiment in Southern France during autumn / winter 2019/20. The field experiment deployed a wide range of in-situ and remote sensing instruments and so the fog events are well documented. A conceptual model for fog previously developed for another site (Toledo et al, 2021) is used to interpret the observations and help to understand the physical processes leading to the onset, development and dissipation of fog in each case. The observational data clearly represents a novel and valuable resource for studying fog,  however I have some questions / suggestions regarding the interpretation of the results and the presentation of the work which I feel need addressing before the manuscript can be published.

**Major comments**

1) Case study papers such as this can often be very descriptive – simply documenting what is seen. While there is value in this, more impact usually comes from the interpretation of the results to learn something new about the underlying physical processes or to identify biases in models. The authors have attempted to do this through the use of the conceptual model, but I still felt that this was only partly successful. The description of the four cases takes up much of the paper and includes the quoting of lots of figures in the text. This is quite hard to read, and by describing each case separately it is hard to compare the figures. A lot of the figures are summarised in table 2. I wonder if they need to all be given in the text as well? Shortening the descriptions in the text would help focus on what was different / interesting about each case.

2) The paper talks about the "conceptual model" and how it can be used for nowcasting, but I struggled to make this link. So far as I can see, what is being presented is a series of diagnostics based on some assumptions about the structure of the fog. These can be used to understand what is happening in the fog, but not necessarily to make predictions about when and how the fog will change. I would suggest that i) you make clear this is a diagnostic model and ii) you either explain how this can be used for nowcasting or remove the references to this. The results seem to suggest that the model diagnoses dissipation an hour or so before the observations show it, but it is not clear to me whether this is due to any real predictive power (i.e. detecting precursors of dissipation) or just because of the different definitions of dissipation used. Either way it gives at most an hour's advanced warning of dissipation, which is of some use, but is a rather limited nowcasting tool.

3) What is the new understanding this paper gives us? We know that radiation fog occurs on its own or sometimes as radiation-advection fog. These are nice case studies of these processes, but it's not clear to me what new understanding we get from them. Can you articulate that clearly both in the introduction (i.e. what the aim of the paper is) and in the conclusions?

4) The use of remote sensing does give valuable information on the vertical structure of the atmosphere, but these instruments do have some limitations in terms of vertical resolution and lowest range gate. In particular the lowest range gate for the radar is 37.5m and for the wind profiler it is 40m. Often fog can be shallow than this and so you are likely missing the early stages of the fog and also cases of shallow fog which does not deepen.

5) The lowest range gate and resolution for the microwave radiometer (MWR) are not given. You mention that lower angle scans were done as well as vertical stares in order to improve vertical resolution. It's not clear what has been presented here in the results though. Do they use a combination? The low angle scans improve vertical resolution, but at the expense of spreading the profile out in the horizontal. Is this important given the heterogeneity at the site? The MWR is also

doing an inversion calculation using neural networks to retrieve the temperature and humidity profiles. In my experience this can often smear / smooth out the profiles compared to radiosondes. You can see your profiles are very smooth curves (unrealistically so). This can be particularly important when looking at features like inversions. Can you comment on this in the context of your results? It might be worth mentioning this as a caveat to the reader.

6) Paragraph at lines 140-147. I was a bit confused what was installed where here. Are the Licor IRGAs at 3m and the sonic anemometers at 10m? Why not mounted at the same height so you can get latent as well as sensible heat fluxes and also use the water vapour to correct the sonic temperature to the true temperature? If at different heights, how do you get the temperature from the Licor? The Licor's are good for measuring rapid fluctuations in water vapour, but have a tendency to drift over time so are not necessarily good for measuring absolute values of humidity without regular calibration. Was this done? They also suffer from issues with water on the lens in rain / fog which can impact on the data quality. Was this an issue? So far as I can see you don't actually use the high frequency water vapour measurements anyway? Later on you plot TKE at 3m (figs 4f, 6f, 8f, 10f) which seems inconsistent with having the sonic anemometer at 10m?

7) In figs 4f, 6f, 8f and 10f you plot TKE from both the sonic anemometer at a point and averaged over layers (from the lidar). How comparable are these values given the different sampling intervals and sampling volumes of the two instruments? It might be worth mentioning that these lines are from very different instruments and so might not be directly comparable.

8) Section 2.1.4. I found this section a bit unclear. I appreciate the authors do not want to reproduce the whole conceptual model here, but there needs to be sufficient detail for the reader to understand the results. The split between the paper and the appendix also seemed slightly arbitrary at times. For example, the critical LWP is a key parameter in determining the RLWP, but this is only defined in the appendix.

**Minor comments**

1) Title "SOFOF3D" → "SOFOG3D"

2) Line 189. "becomes" → "has become"

3) Lines 243-244. "The RLWP gives information about the predictability of fog dissipation time at nowcasting range." - I don't agree with this as written. It does not give information about the predictability at all. What it tells you is whether the fog is likely to begin dissipating due to insufficient water vapour to maintain the surface visibility below 1000m.

4) Line 249. "expectation" → "inspection"?

5) Line 255-256. "At the supersite, the LWP observed during that transition is lower than the threshold at SIRTA (LWP > 30 g m-2)". Why might this be the case? Is it differences in the topography affecting the depth of the fog layer or is it differences in the processes causing fog in this region?

6) Line 263 "triggered" → "were triggered"

7) Line 314. How do you calculate a deep and strong inversion of 14°C km$^{-1}$? I cannot see that big a temperature difference in the figures. This figure is quoted elsewhere in the paper too.

8) Line 316. How do you know this is a low-level jet? The wind field plotted show wind increasing with height. There is no evidence I can see that this is a jet. It might be, but I don't think the observations show it.

9) Line 323. "very low radiative cooling rate". You don't actually measure or plot the radiative cooling rate. I assume you just mean the rate at which the temperature decreases and you are assuming this is all due to radiative cooling? Since you only have the SHF at one height you can't rule out there being some flux divergence leading to warming. Just be careful about how you describe this.

10) Line 356-357. This sentence is confusingly worded. What you mean is "At the supersite, in the absence of any cloud above the fog layer, the fog dissipates after sunrise".

11) Line 362. You talk about "thermal turbulence" in several places, but this can actually be referring to two different processes – either turbulence generated at the top of the dense fog layer due to radiative cooling overnight or turbulence generated at the surface due to solar heating after sunrise. I would be a bit more specific when you talk about thermal turbulence which process you mean.

12) Line 390. "Thin fog (71m)". This is a very precise value given the vertical resolution of the remote sensing instrumentation. This comment applies in general through the paper when giving heights of the fog top / cloud top.

13) Line 392. "associated with decrease" → "are associated with a decrease"

14) Line 421. "fog layer with fall" → "fog layer that then fall"

15) Line 422. "gravity" → "size"? It's not the gravity of the snowflakes which is important it is their weight / size leading to a higher fall speed under the action of the Earth's gravity.

16) Line 460. "low intensity of about 3°C". What does this mean? Do you mean the inversion strength?

17) Line 460-461. "from 20:40 UTC and 23:00 UTC" → "from 20:40 UTC to 23:00 UTC" or "between 20:40 UTC and 23:00 UTC".

18) Line 485. "sustainable dissipation". What does this mean? Do you mean a sustained dissipation rather than a temporary increase in visibility?

19) Line 489. "Advected air mass" → "The advected air mass"

20) Line 494. "resulting to the evolution as a stratus" → "resulting in the evolution of the fog into a stratus cloud".

21) Line 507. "ftime" → "time"

22) Line 507. "such as in" → "as in"

23) Line 518-520. How do the the occurrence of middle and high clouds allow the identification of this as a radiative-advective fog case? I don't follow the logic in this sentence.

24) Line 532-533. "Therefore, …" How does this sentence follow on from the previous sentence? I don't follow the logic.

25) Line 617. "04:00 and 06:40" → "04:00 and 06:40 hours". Same on line 619.

26) Line 623. "fog adiabatic by closure parameter" → "fog adiabatic closure parameter"

27) Line 672-676. Not sure I agree. You have shown these instruments are useful to understand the processes. You've not demonstrated how they help with nowcasting, or indeed that you would need all of them for that. It's a lot to install at a site for nowcasting.

28) Table 2. "Fog top height (FTH)". You only use this phrase / abbreviation in the table. In the text you talk about CTH. Be consistent.

---

## Author Comment (AC1)

**Dear Editor-in-Chief,**

please find in this document our responses to the reviewer #1 of our manuscript.

We thank the reviewer #1 for his/her valuable and constructive suggestions, which led to significant improvements of the quality of our manuscript. Below we detailed how their comments are addressed in the revised version of the manuscript. The corrections made in the manuscript and cited in this document appear in italic.

**Review of 'Role of thermodynamic and turbulence processes on the fog life cycle during SOFOG3D experiment',**

**by Cheikh Dione, Martial Haeffelin, Frederic Burnet, et al.**

**Summary:-**

This paper presents an analysis of data collected during the SOFOG3D campaign in south-west France during 2019-2020. Four cases of fog, and their evolution are discussed with an emphasis on their adiabaticity, and how a conceptual model performs in nowcasting the evolution of each case.

The paper is relevant and interesting, but requires some significant clarification to the data and arguments presented. Some of the analysis presented is difficult to follow, overly complex, and the cause and effect of various processes may be confused. My recommendation is to publish after major revision.

Main points:-

• Description of instrumentation.

Regarding the Doppler Lidar data I would like to see more explanation of how the TKE is retrieved, with an estimate of uncertainty given. I believe that the Lidar must scan over a region of sky to retrieve 3D winds, which raises the likely-hood that air samples in the separate beams are not coherent. Can any independent verification of the calculation be presented here?

We agree with the reviewer on the need of more explanation of how the TKE is retrieved. We added the methodology used to compute the TKE based on Kumer et al., 2016. "*The TKE is computed as the sum of the horizontal velocity variances as in Kumer et al., 2016. Velocity variances are estimated every 30 minutes using the wind components at the high resolution.*" Line 202-204

Kumer, V. M., Reuder, J., Dorninger, M., Zauner, R., Grubišić, V.: Turbulent kinetic energy estimates from profiling wind LiDAR measurements and their potential for wind energy applications, Renew Energy., 99, 898-910, https://doi.org/10.1016/j.renene.2016.07.014, 2016.

Secondly, since the Lidar beam is highly attenuated by liquid water, how much of the fog layer is actually sampled? The authors state between 40 and 220 m, but I believe this is the range of the lidar and not how far the Lidar can typically see into the fog? Other similar Lidars typically can see into around 100m of fog.

We gave the vertical range of the Lidar [40 - 220 m] as an manufactured information. This range can be reach in clear sky (see Figs. 4d, 6d, 8d and 10d). We are aware that the range of the Lidar is reduced by the liquid water content of the fog. This is visible on the wind height cross-section in figures (4d, 6d, 8d and 10d) and the discontinuity of the TKE time series in the 100-400 m layer (Fig 4f, 6f, 8f, and 10f). We further clarified the impact of fog on the vertical resolution of the Wind Lidar. "The wind component are estimated every 10 minutes using a Carrier-to-Noise Ratio (CNR) at least -23 dB and a total data availability of at least 50 %. Note that the CNR depends on atmospheric turbulence characteristics and relative humidity (Aitken et al., 2012). In the presence of fog or low stratus, the Lidar vertical range become low. The TKE is computed as the sum of the horizontal velocity variances as in Kumer et al., 2016. Velocity variances are estimated every 30 minutes using the wind components at the high resolution." Line 198-204

Regarding the Microwave radiometer, an uncertainty is given for absolute humidity, but not the LWP, which is the quantity presented in the figures. Please provide an uncertainty for LWP.

We agreed with the reviewer to add the uncertainty in the retrieval of the LWP. We added in the manuscript the followed sentences "*Martinet et al., 2022 showed that the LWP accuracy has been validated in clear-sky conditions and shown errors between 1 and 14 g m*-2. *These error range is in the scope of that defined in the literature (Crewell and Löhnert, 2003; Marke et al., 2016).*" Line 184-186

Whilst the temperature error of the MWR is quoted as 0.5 degrees for the region of interest, it is clear that the profiles appear highly smoothed in the vertical (compared to what we expect to see from e.g. a tethered balloon profile). This might lead to erroneous conclusions regarding stability and phase of the fog. Were other sources of temperature profiles explored, such as radiosonde, mast or tethered balloon, before using the MWR data? It would be clearer if only the lowest 300m were plotted in the MWR temperature profile plots, and also if fog top were indicated on them at each time.

At the supersite, there was a tethered balloon with varying vertical ranges and temporal resolutions. Note that the temperature measured by this instrument are sensitive to water droplets deposition on the sensors. A few radiosoundings have also been launched at that site but their temporal resolution (3 hours) does not allow to characterize the different fog phases. The only instrument allowing to have a continue estimate of the atmospheric stability remains the MWR.

• Data analysis. Below are examples where I found the presentation of results requires clarification.

L319: - 'thermal turbulence': you mean, 'thermally driven turbulence'?

We mean by thermal turbulence, the turbulence generated by heating. We corrected in the reviewed version of the manuscript. "*The fog dissipation phase is induced by the increase of the vertical mixing generated by the thermal (solar heating) and mechanical turbulence associated with TKE values larger than 0.4 \text{ m}^2 \text{ s}^{-2} (Fig. 4f)." Line 377-379*

L389::- It looks from the figure showing the MWR data that fog starts becoming adiabatic from 2400 or 0100 hours which is inconsistent with the statement here?

We agreed with the review that for these case study, fog starts becoming adiabatic from 00:00 UTC as indicated in the figures illustrating this IOP. We reworded this part of the manuscript following the suggestion of the reviewer #2.

L393:- According to figure 6f sigma w^2 reaches 0.04 by 0300 hours, much more than the figure quoted.

 $\sigma_w^2$  reaches 0.04 m2 s-2 during the fog adiabatic phase. In line 392, we explained the processes driving the fog stable phase. The values indicated refer to Table 2 of the manuscript.

L418-423:- I doubt the conclusion made here. The assertion is that a phase change in the fog caused a reduction in observed LWP, the evidence being 'frost' seen on the balloon cable. It is common to see ice and rime on such things when temperatures are below freezing in fog due to contact-freezing, but this is not evidence of ice or snow in the fog itself. Ice does not generally form in clouds until temperatures become much lower than seen here.

We understand the reviewer's doubts about the explanation we gave for the sudden loss of liquid water of the fog. Considering the evolution of the LWP, near surface temperature, the observations reported by the scientists operating on the supersite and weather reports from Météo-France at Biscarrosse and Bergerac located western (~60 km) and northeastern (~100 km) the supersite (https://www.ogimet.com/cgi-bin/gsynres?

lang=en&ind=07503&decoded=yes&ndays=2&ano=2020&mes=01&day=06&hora=12, https://www.ogimet.com/cgi-bin/gsynres?

lang=en&ind=07530&decoded=yes&ndays=2&ano=2020&mes=01&day=06&hora=12), we have explained this process by the formation of a freezing fog. We reworded this part following the suggestion of the reviewer #2. Line 399-403

Section 3.3:- generally difficult to follow. Examples below.

L455:- Fig. 8a indicates a weak inversion at 2100 before the stratus lowers into the fog.

Following the suggestion of the reviewer #2, we reworded this part of the manuscript by proposing a short description of the case studies 2, 3 and 4.

L458-459:- How does slowing down the cooling create a thin layer of temperature inversion?

We means "slowing down the wind speed" instead of "slowing down the cooling rate". We agreed with the reviewer that this sentence is unclear. We reworded this part of the manuscript following the suggestion of the reviewer #2.

L476:- Turbulence levels are very low for this case so how was the transition driven by turbulence?

The transition is driven by mechanical turbulence generated by the brisk winds (wind shear) observed at around 23:30 UTC. It is after that time that the turbulence started to decrease.

L458:- What is 'sustainable dissipation'?

We mean by "sustainable dissipation" " *a definitive dissipation of fog*". We corrected the review version of manuscript. Line 450

L489. Why would a warming allow a deepening of the fog layer? I suggest the dissipation of this layer can be more simply put: Increasing wind aloft brought warm drier air over the top of the fog

that then mixed into it, evaporating fog droplets, reducing RLWP to negative values and causing the fog to lift into low stratus.

We accepted reviewer's suggestions and reworded this part of the manuscript as "Increasing wind aloft brings warm drier air over the top of the fog that then mixing into it (TKE =  $0.33 \text{ m}^2 \text{ s}^{-2}$  and  $\sigma_w^2 = 0.07 \text{ m}^2 \text{ s}^{-2}$ ), evaporating fog droplets, reducing the RLWP to negative values and causing the fog to lift into low stratus. The fog dissipation phase is thus driven by the advection of warm air at the supersite. "Line 427-431

L518:- 'triggering of the ultra-low stratus being the fog'. Why not just say this was 'stratus fog'?

Suggestion accepted. Line 477

L576:- Low stratus is not fog.

We corrected in the reviewed version of the manuscript as "*The combination of advection and radiative cooling favours stratus fog formation at about 150 m a.g.l followed by a rapid (less than 30 min) lowering of it base height to the surface triggering the onset of the fog in an unstable (case 3) and neutral (case 4) surface atmospheric boundary layer (Fig. 11c and 11d)*". Line 476-479

---

## Author Comment (AC2)

**Dear Editor-in-Chief,**

please find in this document our responses to the reviewer #2 of our manuscript.

We thank the reviewer #1 for his/her valuable and constructive suggestions, which led to significant improvements of the quality of our manuscript. Below we detailed how their comments are addressed in the revised version of the manuscript. The corrections made in the manuscript and cited in this document appear in italic.

Review of egusphere-2023-1224 : Role of thermodynamic and turbulence processes on the fog life cycle during SOFOF3D

This manuscript presents case studies of four of the heaviest fog events from the SOFOG3D field experiment in Southern France during autumn / winter 2019/20. The field experiment deployed a wide range of in-situ and remote sensing instruments and so the fog events are well documented. A conceptual model for fog previously developed for another site (Toledo et al, 2021) is used to interpret the observations and help to understand the physical processes leading to the onset, development and dissipation of fog in each case. The observational data clearly represents a novel and valuable resource for studying fog, however I have some questions / suggestions regarding the interpretation of the results and the presentation of the work which I feel need addressing before the manuscript can be published.

**Major comments**

1) Case study papers such as this can often be very descriptive – simply documenting what is seen. While there is value in this, more impact usually comes from the interpretation of the results to learn something new about the underlying physical processes or to identify biases in models. The authors have attempted to do this through the use of the conceptual model, but I still felt that this was only partly successful. The description of the four cases takes up much of the paper and includes the quoting of lots of figures in the text. This is quite hard to read, and by describing each case separately it is hard to compare the figures. A lot of the figures are summarised in table 2. I wonder if they need to all be given in the text as well? Shortening the descriptions in the text would help focus on what was different / interesting about each case.

We agree with the reviewer that the description of the four case studies is quite long. This is explained by the fact that the objective of this paper is to properly document the physical processes driving each fog phase during it evolution and for each case study. Each case study is associated with specific processes driving the evolution of the fog phases. However, we have reduced the description of the case studies and improve the quality of the text. In the review version of the manuscript, we give a detailed analysis of the case study 1 and shortened the three other cases.

2) The paper talks about the "conceptual model" and how it can be used for nowcasting, but I struggled to make this link. So far as I can see, what is being presented is a series of diagnostics based on some assumptions about the structure of the fog. These can be used to understand what is happening in the fog, but not necessarily to make predictions about when and how the fog will change. I would suggest that i) you make clear this is a diagnostic model and ii) you either explain how this can be used for nowcasting or remove the references to this. The results seem to suggest that the model diagnoses dissipation an hour or so before the observations show it, but it is not clear to me whether this is due to any real predictive power (i.e. detecting precursors of dissipation) or just because of the different definitions of dissipation used. Either way it gives at most an hour's advanced warning of dissipation, which is of some use, but is a rather limited nowcasting tool.

We agree with the reviewer that the conceptual model used in this paper serve as a diagnostic for the understanding of fog evolution phases. When the fog is quite developed, the model allows the monitoring of the different fog phases. It is not a numerical weather forecasting model that can be used for nowcasting but rather a basic tool that can be used in nowcasting. In the new version of the manuscript, we presented the model as a nowcasting tool.

3) What is the new understanding this paper gives us? We know that radiation fog occurs on its own or sometimes as radiation-advection fog. These are nice case studies of these processes, but it's not clear to me what new understanding we get from them. Can you articulate that clearly both in the introduction (i.e. what the aim of the paper is) and in the conclusions?

This study is the first documenting fog thermodynamical processes driving the fog evolution phases over the southwestern France. It highlights the main physical processes (advection, mechanical and thermal turbulence) driving fog evolution.

We have reviewed in the introduction and conclusions of the manuscript, the parts presenting the objective of this paper. The following modifications are added in the introduction. "In particular, the role of horizontal advection, atmospheric stability and turbulence is further analyzed to better identify the drivers of fog phases. "(Lines 89-90) and in the conclusions section "Based on an innovative instrumental synergy combining in-situ and remote sensing measurements gathered in an adiabatic fog conceptual model, this study has documented the dynamical (advection) and thermodynamical (atmospheric stability, turbulence) processes favoring fog formation, evolution and dissipation of two categories of fog (radiation and radiation-advection) over Southwestern France." Line 543-548

4) The use of remote sensing does give valuable information on the vertical structure of the atmosphere, but these instruments do have some limitations in terms of vertical resolution and lowest range gate. In particular the lowest range gate for the radar is 37.5m and for the wind profiler it is 40m. Often fog can be shallow than this and so you are likely missing the early stages of the fog and also cases of shallow fog which does not deepen.

We are aware that the remote sensing used in this paper has limitations on the vertical resolution. Although the radar does not allow to see with precision the very thin fogs (CTH  $\leq$  37.5 m), we are able to define fog onset using the horizontal visibility at near surface.

To complete the information from the wind lidar, we used the TKE at 3 m from the mast. In addition, an instrumented mast on several levels up to 50 m a.g.l was installed at the supersite at the start of the campaign. Unfortunately, it was destroyed by the storm Amélie on November 3, 2019. For the rest of the campaign, only the firsts levels (2 m and 3 m) were maintained. The remote sensing information is valuable when the fog thickness is well developed vertically.

5) The lowest range gate and resolution for the microwave radiometer (MWR) are not given. You mention that lower angle scans were done as well as vertical stares in order to improve vertical resolution. It's not clear what has been presented here in the results though. Do they use a combination? The low angle scans improve vertical resolution, but at the expense of spreading the profile out in the horizontal. Is this important given the heterogeneity at the site? The MWR is also doing an inversion calculation using neural networks to retrieve the temperature and humidity profiles. In my experience this can often smear / smooth out the profiles compared to radiosondes. You can see your profiles are very smooth curves (unrealistically so). This can be particularly important when looking at features like inversions. Can you comment on this in the context of your results? It might be worth mentioning this as a cave at to the reader.

As MWRs are passive instruments, there is no real limitation in the vertical. However, that is true that, depending on the weighting functions, the information content of the instrument can be

degraded depending on the altitude. For channels used for temperature profiling, most of the information content is located within the first 4 km of altitude, most of the information being contained within the first kilometer. The information content of the instrument can be increased by adding measurements at several elevation angles from approximately 5° above ground in addition to the zenith angle. Temperature profiles shown in this study were retrieved from the MWR measurements made at all elevation angles (the lower elevations angles added to measurements at zenith). That is true that, to be rigorous, low elevation angles should be used when the atmosphere is supposed to be homogeneous in the horizontal at a distance of approximately 1 km (in which the first 500 meters are the most important) around the instrument. Several studies have demonstrated that, even in complex terrain in which this assumption is violated, boundary layer scans can still improve the temperature profile accuracy within the first kilometer compared to radiosoundings (Martinet et al, 2017). In this study, the homogeneity assumption should be respected as the terrain around the MWR is completely flat with no surface heterogeneity (it is entirely representative of cultivated fields). The addition of boundary layer scans for this study is thus beneficial in regards to the improved vertical resolution compared to using only zenith measurements.

When boundary layer scans are used, the expected vertical resolution of the instrument is around 50 to 100m for the first gates. The vertical resolution then degrades with altitude up to 500m at 1km. Temperature profiles retrieved from MWRs are thus smooth in line with the spread of the weighting functions. However, MWRs profiles are used in this study only to identify the evolution of fog between formation, stable, unstable and dissipation phases during well developed fog cases. To that end, the main proxy is the temperature gradient within the first 100 meters and the temporal evolution of this gradient which is well resolved by the MWR.

**References**

Martinet, P., Cimini, D., De Angelis, F., Canut, G., Unger, V., Guillot, R., Tzanos, D., and Paci, A.: Combining ground-based microwave radiometer and the AROME convective scale model through 1DVAR retrievals in complex terrain: an Alpine valley case study, Atmos. Meas. Tech., 10, 3385–3402, https://doi.org/10.5194/amt-10-3385-2017, 2017.

6) Paragraph at lines 140-147. I was a bit confused what was installed where here. Are the Licor IRGAs at 3m and the sonic anemometers at 10m? Why not mounted at the same height so you can get latent as well as sensible heat fluxes and also use the water vapour to correct the sonic temperature to the true temperature? If at different heights, how do you get the temperature from the Licor? The Licor's are good for measuring rapid fluctuations in water vapour, but have a tendency to drift over time so are not necessarily good for measuring absolute values of humidity without regular calibration. Was this done? They also suffer from issues with water on the lens in rain / fog which can impact on the data quality. Was this an issue? So far as I can see you don't actually use the high frequency water vapour measurements anyway? Later on you plot TKE at 3m (figs 4f, 6f, 8f, 10f) which seems inconsistent with having the sonic anemometer at 10m?

We agree with the review that as written in the paper there is a bit confusion on the installation of the Licor and the sonic anemometer. These two instruments were installed at the same level (3 m a.g.l). Yes, the Licor was calibrated according to the manufacturer's specifications. This calibration was done 15 days before it installation. The humidity problem on the Licor has not been resolved. We haven't data when the automatic gain control (AGC of the sensor is too high meaning the presence of large droplets. This installation was completed by a slow Vaisala HMP110 humidity senso to obtain an absolute value.

We corrected on the reviewed version of the paper as "Meteo-France installed in a fallow field near the supersite, several sensors as Licor analyzers and sonic anemometers to continuously measure the near-surface (3 m a.g.l) meteorological conditions (air temperature and relative humidity), the three components of the wind, and pressure at 0.3 m a.g.l). These instruments provided high frequency data at 20 Hz. In this study, to document turbulence and thermodynamical processes

driving fog phases, we use sensible heat flux (SHF), turbulence kinetic energy (TKE), and vertical velocity variance ( $\sigma_w^2$ ). These variables are estimated using the Eddy-covariance methods (Foken et al., 2004, Mauder et al., 2013) calculated every 30 minutes after a high quality control of the data. More details on the data can be found in Canut, 2020. "Lines 138-145.

7) In figs 4f, 6f, 8f and 10f you plot TKE from both the sonic anemometer at a point and averaged over layers (from the lidar). How comparable are these values given the different sampling intervals and sampling volumes of the two instruments? It might be worth mentioning that these lines are from very different instruments and so might not be directly comparable.

Our objective isn't to compare the two TKE estimates but rather to see the consistency of the two instruments in the estimate of turbulence considering the same time interval (30 minutes) to see the vertical evolution of the turbulence by combining the information from these two instruments. Their consistency provides information on the quantification of turbulence from near surface up to 220 m a.g.l. Based on the temporal resolution used for the restitution of the TKE and from light to moderate turbulence, the two instruments should approximately detect the same eddies although the vertical resolution of the lidar (20 m) larger then the sonic anemometer. This is what we observed in figs 4f, 6f, 8f and 10f.

8) Section 2.1.4. I found this section a bit unclear. I appreciate the authors do not want to reproduce the whole conceptual model here, but there needs to be sufficient detail for the reader to understand the results. The split between the paper and the appendix also seemed slightly arbitrary at times. For example, the critical LWP is a key parameter in determining the RLWP, but this is only defined in the appendix.

For a good understanding of the methodology, we accept the reviewer's suggestion to put the critical LWP estimation method in the paper instead of in the appendix. In the reviewed version of the manuscript, we added the paragraph A4 after equation 2. Lines 241-252.

**Minor comments**

1) Title "SOFOF3D" → "SOFOG3D"

This has been corrected. Line 2

2) Line 189. "becomes"  $\rightarrow$  "has become"

This has been corrected. Line 189

3) Lines 243-244. "The RLWP gives information about the predictability of fog dissipation time at nowcasting range." - I don't agree with this as written. It does not give information about the predictability at all. What it tells you is whether the fog is likely to begin dissipating due to insufficient water vapour to maintain the surface visibility below 1000m.

The RLWP provides an estimation of the excess/deficit of water of fog in near real time that enables the fog layer to remain at the surface or dissipate. It can be used as a diagnostic to estimate how likely fog persistence is for the coming minutes, hours, based on its instantaneous values and its trend. This really conduct to fog dissipation nowcasting tool. We reworded this sentence for a further understanding of the used of the RLWP in this study as "*The RLWP gives an estimation of the excess/deficit of liquid water of the fog that enables the fog layer to remain at surface or dissipate. It can be used as a diagnostic of how likely the fog will persist in the coming minutes, hours (nowcasting of fog dissipation time).*" Line 258-261

4) Line 249. "expectation"  $\rightarrow$  "inspection"?

Yes, we mean 'inspection'. Line 266

5) Line 255-256. "At the supersite, the LWP observed during that transition is lower than the threshold at SIRTA (LWP > 30 g m-2)". Why might this be the case? Is it differences in the

topography affecting the depth of the fog layer or is it differences in the processes causing fog in this region?

This difference could be explained by the environments surrounding the two sites. The SIRTA is a peri-urban site under the influence of pollution from road traffic, heating etc. This pollution can contribute to the load of the atmospheric boundary layer in condensation nuclei favoring thick fogs with (visibility < 100 m). "La Charbonnière" is a rural site less polluted and under the influence of the Atlantic Ocean, mountains and lands forests.

6) Line 263 "triggered" → "were triggered" This has been corrected. Line 280

7) Line 314. How do you calculate a deep and strong inversion of 14°C km -1 ? I cannot see that big a temperature difference in the figures. This figure is quoted elsewhere in the paper too.

The inversion is calculated base the difference between temperatures at the base and top heights. The base is defined as the level that the air temperature start to increase and the top corresponds to the level that the temperature start again to decrease. We corrected in the reviewed version of the manuscript. The inversion magnitude is around 8  $^{\circ}$ C km-1. Line 332.

8) Line 316. How do you know this is a low-level jet? The wind field plotted show wind increasing with height. There is no evidence I can see that this is a jet. It might be, but I don't think the observations show it.

Based on identical wind configuration between the IOPs 5 and 6 from the wind Lidar, we look at the radiosoundings of IOP 6. The vertical profiles of wind observed before fog onset show a low-level jet (see figure 1 below) on January 5, 2020 at 18:00 and 21:43 UTC. The jet core height is at around 500 m a.g.l. We based on this case study to assert that there is indeed a low-level jet for IOP 5. The same observations are also observed for IOP 11 and 14.

*Figure 1:* Vertical profiles of wind speed (left) and wind direction (right) from the radiosoundings lunched at the supersite during IOP6. The 4 first digit number of the legend indicate the date in MMDD and the last 4 digit number the time in hhmm. MM for month, DD for day, hh for hour and mm for minute. The times are in UTC.

9) Line 323. "very low radiative cooling rate". You don't actually measure or plot the radiative cooling rate. I assume you just mean the rate at which the temperature decreases and you are

assuming this is all due to radiative cooling? Since you only have the SHF at one height you can't rule out there being some flux divergence leading to warming. Just be careful about how you describe this.

We mean "cooling rate" instead of "radiative cooling rate". We corrected in the reviewed version of the manuscript. "*low negative near surface cooling rate* ". Line 340

10) Line 356-357. This sentence is confusingly worded. What you mean is "At the supersite, in the absence of any cloud above the fog layer, the fog dissipates after sunrise".

We accepted the suggestion and modified this sentence in the reviewed version of the manuscript "*At the supersite, in the absence of any cloud above the fog layer, the fog dissipates after sunrise.*". Line 371-372

11) Line 362. You talk about "thermal turbulence" in several places, but this can actually be referring to two different processes – either turbulence generated at the top of the dense fog layer due to radiative cooling overnight or turbulence generated at the surface due to solar heating after sunrise. I would be a bit more specific when you talk about thermal turbulence which process you mean.

Here, we mean thermal turbulence by the turbulence generated by the solar heating after sunrise. We reworded this sentence as "*The fog dissipation phase is induced by the increase of the vertical mixing generated by the thermal (solar heating) and mechanical turbulence associated with TKE values larger than 0.4 \text{ m}^2 \text{ s}^{-2} (Fig. 4f)." Line 376-378*

12) Line 390. "Thin fog (71m)". This is a very precise value given the vertical resolution of the remote sensing instrumentation. This comment applies in general through the paper when giving heights of the fog top / cloud top.

All fog thickness values given in the text were averaged in a time range. We agree that 71 m is not in the range of the Radar vertical resolution. We corrected taking the Radar closest gate to the CTH mean computed.

13) Line 392. "associated with decrease"  $\rightarrow$  "are associated with a decrease"

We reworded this part of the manuscript following the first suggestion of the reviewer.

14) Line 421. "fog layer with fall"  $\rightarrow$  "fog layer that then fall"

We reworded this part based on the suggestion of reviewer 1 and your suggestion for shortening the description of the IOPs. Line 402-403

15) Line 422. "gravity"  $\rightarrow$  "size"? It's not the gravity of the snowflakes which is important it is their weight / size leading to a higher fall speed under the action of the Earth's gravity.

Yes, we mean the size instead of gravity. Thank. Line 402-403

16) Line 460. "low intensity of about 3°C". What does this mean? Do you mean the inversion strength?

Here, we means the temperature inversion strength. Following your first suggestion in major comments section, we reworded this part in the reviewed version of the manuscript.

17) Line 460-461. "from 20:40 UTC and 23:00 UTC"  $\rightarrow$  "from 20:40 UTC to 23:00 UTC" or "between 20:40 UTC and 23:00 UTC".

We mean "from 20:40 UTC to 23:00 UTC" but this paragraph is reworded in the reviewed version of the manuscript as you suggested in your first suggestion in major comments section.

18) Line 485. "sustainable dissipation". What does this mean? Do you mean a sustained dissipation rather than a temporary increase in visibility?

In the previous version of the manuscript, we mean by "sustainable dissipation" a "*definitive dissipation*" of the fog. Line 436

19) Line 489. "Advected air mass"  $\rightarrow$  "The advected air mass"

We corrected this part following the suggestions of the two reviewers. "Increasing wind aloft brings warm drier air over the top of the fog that then mixed into it (TKE =  $0.33 \text{ m}^2 \text{ s}^{-2}$  and  $\sigma_w^2 = 0.07 \text{ m}^2 \text{ s}^{-2}$ ), evaporating fog droplets, reducing the RLWP to negative values and causing the fog to lift into low stratus. The fog dissipation phase is thus driven by the advection of warm air at the supersite (Fig. 7e)." Line 426-430

20) Line 494. "resulting to the evolution as a stratus"  $\rightarrow$  "resulting in the evolution of the fog into a stratus cloud".

This has been corrected in the previous comment.

21) Line 507. "ftime" → "time"

We reworded this part of the manuscript following the first suggestion in major comments section of the reviewer.

22) Line 507. "such as in"  $\rightarrow$  "as in"

We reworded this part of the manuscript following the first suggestion in major comments section of the reviewer.

23) Line 518-520. How do the the occurrence of middle and high clouds allow the identification of this as a radiative-advective fog case? I don't follow the logic in this sentence.

We known that middle clouds are associated with a change in weather (advection of air mass). They are observed at the supersite when the stratus fog formed.

24) Line 532-533. "Therefore, ..." How does this sentence follow on from the previous sentence? I don't follow the logic.

We reworded this part of the manuscript following the first suggestion in major comments section of the reviewer.

25) Line 617. "04:00 and 06:40" → "04:00 and 06:40 hours". Same on line 619.

This has been corrected. Line 516 and 518

26) Line 623. "fog adiabatic by closure parameter" → "fog adiabatic closure parameter"

This has been corrected. Line 522

27) Line 672-676. Not sure I agree. You have shown these instruments are useful to understand the processes. You've not demonstrated how they help with nowcasting, or indeed that you would need all of them for that. It's a lot to install at a site for nowcasting.

We reworded this part by highlighting the usefulness of having these instruments to monitor fog in near real-time. Monitoring fog is the basis of it nowcasting. This is the added-value of how these instruments can help to nowcasting fog.

We reworded this part as

"Indeed, this paper highlights that fog nowcasting tools in this region needs in addition to the numerical weather prediction models, a cloud radar, a microwave radiometer, a wind lidar, a surface energy balance, and meteorological stations. Operationalizing these instruments would allow to improve fog nowcasting, which will reduce its socioeconomic impacts in this region." Line 590-594

28) Table 2. "Fog top height (FTH)". You only use this phrase / abbreviation in the table. In the text you talk about CTH. Be consistent.

This has been corrected in the new version of the manuscript. We used CTH (cloud top height). See Table 2 and line 874.

---

## Referee Report (RR1)

**Review of egusphere-2023-1224, revision 2**

This is a much improved revision of the manuscript taking into account the comments of both reviewers. In particular the description of the case studies is more concise and the similarities and differences between the different cases is more clearly articulated. There is also an attempt to better explain the rationale for using the conceptual model. I still feel that while this is a very nicely documented case study it is somewhat incremental in terms of developing our scientific understanding. There is however sufficient novelty to merit publication. The paper does demonstrate that there could be some utility in using the conceptual model for very short range nowcasting. In very specific situations (e.g. aviation) the impact of fog and the need for advanced warning of fog dissipation might justify the cost of the instrumentation required. I have only some minor technical comments now, and once these are addressed, I am happy to recommend publication of the manuscript.

(Line numbers correspond to the track changes version submitted by the authors).

Line 137. "Meteo-France installed in a fallow field near the supersite, several sensors as Licor analyzers and sonic anemometers" -> "At a fallow field near the supersite Meteo France installed a sonic anemometer and Licor infra-red gas analyzer (IRGA)". Do you need to mention the Licor? You don't actually use that.

Line 261. "will persist in the coming minutes, hours (nowcasting tool for fog dissipation time)." -> "will persist in the coming minutes and hours (i.e. as a nowcasting tool for fog dissipation time)."

Line 335. "could be considered as a low-level jet". I raised this in my previous review, and the response is not entirely satisfactory. It *might* be a jet, but the observations presented don't actually show that. I would suggest some thing like "possibly due to a low-level jet" instead to make it clear that this is something you are hypothesising by analogy with IOP6 rather than something you know.

Line 345. "an horizontal visibility" -> "a horizontal visibility"

Line 373. "a continue increase in SHF" -> "a continued increase in SHF"

Line 443. "Fog adiabatic phase" -> "The fog adiabatic phase"